# Aligning Gradient and Hessian for Neural Signed Distance Function

**Ruian Wang**[*]
Shandong University
`wra.time@gmail.com`

**Zixiong Wang**[*]
Nankai University
`zixiong_wang@outlook.com`

**Yunxiao Zhang**
Shandong University
`zhangyunxiaox@gmail.com`

**Shuangmin Chen**[†]
Qingdao University of
Science and Technology
`csmqq@163.com`

**Shiqing Xin**
ShanDong University
`xinshiqing@sdu.edu.cn`

**Changhe Tu**
Shandong University
`chtu@sdu.edu.cn`

**Wenping Wang**
TEXAS A&M UNIVERSITY
`wenping@cs.hku.hk`

## Abstract

The Signed Distance Function (SDF), as an implicit surface representation, provides a crucial method for reconstructing a watertight surface from unorganized point clouds. The SDF has a fundamental relationship with the principles of surface vector calculus. Given a smooth surface, there exists a thin-shell space in which the SDF is differentiable everywhere such that the gradient of the SDF is an eigenvector of its Hessian matrix, with a corresponding eigenvalue of zero. In this paper, we introduce a method to directly learn the SDF from point clouds in the absence of normals. Our motivation is grounded in a fundamental observation: aligning the gradient and the Hessian of the SDF provides a more efficient mechanism to govern gradient directions. This, in turn, ensures that gradient changes more accurately reflect the true underlying variations in shape. Extensive experimental results demonstrate its ability to accurately recover the underlying shape while effectively suppressing the presence of ghost geometry.

## 1 Introduction

In recent years, the neural signed distance function (SDF) has demonstrated its capability to represent high-fidelity geometry [37, 16, 8, 53, 1, 19]. Existing approaches primarily employ neural networks to map coordinates to their corresponding signed distance values. Depending on whether or not supervision is used, these approaches can be classified into two categories: learning-based methods and optimization-based methods, where the former fits data samples to their corresponding ground-truth implicit representations [37, 16, 8, 22, 21] and the latter directly infer the underlying SDF from point clouds [1, 19, 5, 3, 48] or multi-image [45, 29, 35]. Despite significant advancements in SDF-based surface reconstruction, both types of methods have their drawbacks. The supervised reconstruction methods may not generalize well [42] on shapes or point distributions that are not present in the training data. The optimizing-based methods, on the other hand, struggle to resolve the ambiguity of the input point cloud, particularly when normal information is absent. In this paper, our

---

[*]These authors contributed equally.
[†]S. Chen is the corresponding author.

37th Conference on Neural Information Processing Systems (NeurIPS 2023).

focus is primarily on optimization-based reconstruction techniques for point clouds without normal information.

The majority of optimization-based approaches utilize first-order constraints to regulate the variation of the SDF. For example, the Eikonal term [19, 40, 53] is commonly employed to ensure that the gradients are unit vectors. However, as pointed out in [6], the Eikonal term is weak in its ability to regulate the direction of the gradients. As a result, it is challenging to prevent the emergence of ghost geometry (the level sets of the SDF are highly disordered) and unnecessary shape variations even when the Eikonal term is enforced. To address these issues, various second-order smoothness energy formulations [54, 6] are proposed to steer the direction of the gradients to change toward a desirable configuration. However, it entails considerable difficulty to regulate the extent to which the smoothness term is enforced. A commonly seen artifact is that the resulting surface may be excessively smooth.

In this research, we re-examine the problem of surface reconstruction based on surface vector calculus, and introduce a novel loss to facilitate the inference of a faithful SDF directly from raw point clouds without oriented normals. An interesting observation is that within the narrow thin-shell space of the underlying surface, where the real SDF is differentiable everywhere in the narrow thin-shell space, the gradient of the SDF is an eigenvector of its Hessian matrix and the corresponding eigenvalue is zero. Specially, when a point is situated on a surface, the normal vector transforms into an eigenvector of the Hessian of the SDF, with the corresponding eigenvalue being zero. In essence, the gradient and Hessian of the SDF must be aligned within a thin-shell region surrounding the underlying surface. This alignment allows for more effective control over the direction of the gradients. Building on the key observation, we develop a new loss function that promotes the alignment of the gradient and Hessian of the SDF, rather than relying on smoothness energy. We have conducted a comprehensive evaluation of our proposed approach on a variety of benchmarks and compared it to recently proposed reconstruction methods. Extensive experimental results show that our approach outperforms the state-of-the-art techniques, whether overfitting a single shape or learning a shape space. It can not only accurately recover the underlying shape but also suppress the occurrence of ghost geometry, which verifies the effectiveness of the gradient-Hessian alignment.

## 2 Related Work

### 2.1 Traditional Methods

Traditional reconstruction methods can be classified into two categories: explicit and implicit. Explicit methods focus on establishing direct connections between input points, while implicit methods aim to fit an implicit field that conforms to the given points and normals. Roughly speaking, popular explicit reconstruction techniques utilize computational geometry methods, such as Delaunay triangulation or Voronoi diagrams, to infer connections between points. These methods can produce well-tessellated triangle mesh surfaces [26, 14, 47]. However, they may struggle to ensure manifoldness, particularly when the input point cloud contains defects such as irregular point distribution, noise, or missing parts. In contrast, implicit reconstruction techniques [25, 23, 39] are capable of generating manifold and watertight surfaces. However, the majority of these methods necessitate that the input point cloud is equipped with normal information.

In the event that the provided point cloud is devoid of normals, it is necessary to either estimate the normals and orientations prior [52] to surface reconstruction or devise a novel reconstruction framework that can operate in the absence of normal information [20, 27]. In essence, all of them have to estimate normals either prior to or during the process of surface reconstruction. As a result, these approaches remain heavily reliant on the quality of the estimated normals. In cases where the provided point cloud is of poor quality, it becomes challenging for these methods to overcome the inherent ambiguity introduced by the absence of normal information.

### 2.2 Supervision-based Learning Methods

Learning-based methodologies have demonstrated a superior capacity for reconstruction, particularly in the context of neural implicit function [51]. They optimize the network to implicitly encode a signed distance field or occupancy field with the supervision of ground truth. Preceding methodologies primarily employ object-level priors [37, 34, 18] to encode shapes, which are referred to as global

priors. However, the utilization of global priors restricts the generation ability as the network is unable to deal with shapes that are absent from the training set. Subsequently, numerous methodologies concentrate on local data priors [22, 16, 8, 21] to enhance generalization capacity by utilizing small receptive fields. The approaches for acquiring local data priors encompass regular grid [22], KNN [17, 16, 8], and octree [46, 44] and so on. Although local data priors enhance the generation ability of learning-based methodologies, the priors derived from limited data may result in excessively smoothed surfaces. At the same time, the reconstruction performance may further deteriorate if the distribution of the test point cloud significantly deviates from the training data [42].

## 2.3 Optimization-based Learning Methods

To enhance the generalization ability, the direct fitting of 3D representations from raw point clouds without supervision has been extensively investigated in recent years, enabling end-to-end prediction of the target surface. The majority of optimization-based reconstructions necessitate the fitting of individual shapes with specific network parameters, where the parameters are acquired by imposing additional constraints. SAL/SALD [1, 2] employs unsigned distances to facilitate sign-agnostic learning. IGR [19] and Neural-Pull [5] primarily utilize Eikonal terms to constrain the field to be an SDF. Additionally, several methodologies [3, 4, 55] have been modified from Neural-Pull in pursuit of enhanced quality. DiGS [6] incorporates Laplacian energy into neural implicit representation learning for unoriented point clouds. In summary, the preponderance of existing methodologies constrain the norm of gradients or minimize smoothness energy to attain equilibrium between geometric details and overall simplicity, but this inevitably results in ghost geometry or over-smoothed reconstruction outcomes. In this paper, our emphasis is on the alignment of the gradients and the Hessian of the SDF to more effectively regulate the direction of the gradients.

# 3 Method

## 3.1 Neural signed distance function

Given an unoriented point cloud $\boldsymbol{P}$ restricted in the range $\Omega : [-1, 1] \times [-1, 1] \times [-1, 1]$, our task is to find a neural signed distance function (SDF) $f_\theta : \mathbb{R}^3 \mapsto \mathbb{R}$, such that $f_\theta$ predicts a signed distance value for an arbitrary query point $\boldsymbol{q} \in \Omega$, where the neural function $f_\theta$ is parameterized by $\theta$ and can be assumed to be $C^2$-continuous everywhere in the domain. We use $\mathcal{S}_l$ to denote the level-set surface at the value of $l$, i.e.,

$$\mathcal{S}_l = \left\{ \boldsymbol{q} \in \mathbb{R}^3 \mid f_\theta(\boldsymbol{q}) = l \right\}. \tag{1}$$

The underlying surface can be naturally obtained by extracting the zero level-set surface $\mathcal{S}_0$, through the use of contouring algorithms, such as Marching cubes [30]. The task of this paper is to fit the high-fidelity implicit representation without supervision.

## 3.2 Surface vector calculus related to SDF

In order to fit the SDF, there are typically three types of boundary conditions for constraining $f_\theta$: (1) Dirichlet condition $f_\theta(p) = 0$ that requires each point $p \in \boldsymbol{P}$ to be situated on $\mathcal{S}_0$ as far as possible, (2) Eikonal condition $\|\nabla f_\theta\|_2 = 1$ that enforces $f_\theta$ be a distance field with unit gradients, and (3) Neumann condition $\nabla f_\theta = \mathcal{N}$ that aims to align the gradients with the normal field $\mathcal{N}$. Existing neural implicit representations incorporate the aforementioned boundary conditions as constraints, either explicitly [19, 40] or implicitly [5, 1]. However, it is important to note that the Neumann condition cannot be enforced due to the absence of oriented normals.

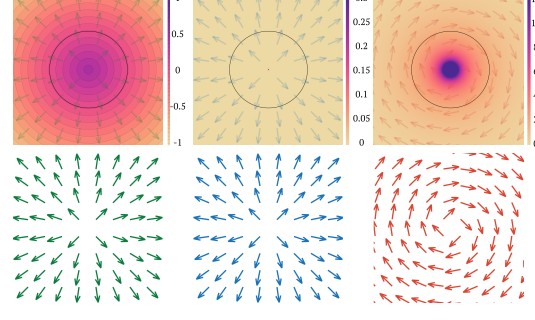

Figure 1: A 2D visual for differential property of SDF for circle. From left to right: SDF with gradient, minimum (0) and maximum eigenvalues of the Hessian matrix with their corresponding eigenvector.

According to the surface vector calculus [32], the SDF has a fundamental relationship with the differential properties of a smooth surface. Let $\mathbf{J}_{\nabla f_\theta}$ denote the Jacobian matrix of the gradient $\nabla f_\theta$,

or equivalently the Hessian matrix $\mathbf{H}_{f_\theta}$ of $f_\theta$. Since this matrix is symmetric and diagonalizable, we can find the eigenvectors and eigenvalues of $\mathbf{H}_{f_\theta}$ for any point where $f_\theta$ is differentiable (a narrow thin-shell space surrounding the base surface). By differentiating both sides of the identity $\|\nabla f_\theta\|_2 = 1$, we get:

$$\mathbf{H}_{f_\theta} \nabla f_\theta(\boldsymbol{q}) = \mathbf{0}, \tag{2}$$

which implies that $\nabla f_\theta(\boldsymbol{q})$ is exactly an eigenvector of the Hessian $\mathbf{H}_{f_\theta}(\boldsymbol{q})$, with a corresponding eigenvalue of 0. To be more detailed, when $\boldsymbol{q}$ lies on the surface, the gradient represents the normal vector at $\boldsymbol{q}$. Consequently, we have

$$\mathbf{H}_{f_\theta} \mathcal{N}(\boldsymbol{q}) = \mathbf{0}. \tag{3}$$

Additionally, the other two eigenvectors of $\mathbf{H}_{f_\theta}(\boldsymbol{q})$ define the principal directions at $\boldsymbol{q}$ [32, 36], with the eigenvalues respectively being the opposite of the corresponding principal curvatures. If $\boldsymbol{q}$ is off the surface but in the differentiable region, the three eigenvectors reflect the differential properties of the closest surface point of $\boldsymbol{q}$, i.e., the projection of $\boldsymbol{q}$ onto the surface. In summary, the eigenvectors of the Hessian can disclose the fundamental properties of the underlying surface.

A similar property holds in the 2D setting. In Fig. 1, we provide a 2D circle as a toy example to illustrate this property. It's evident that in the vicinity of the base surface, the Hessian matrix of the SDF consistently exhibits two eigenvectors: one aligned with the gradient, and the other orthogonal to the gradient. Moreover, we visualize the distribution of the eigenvalues using a color-coded style, with the eigenvalue corresponding to the gradient being 0.

## 3.3 Gradient-Hessian Alignment

It should be noted that even if the underlying surface is infinitely smooth, the SDF may not necessarily be smooth everywhere. From a geometric perspective, the SDF is non-differentiable at medial-axis points with at least two nearest projections onto the surface. The neural implicit function $f_\theta$, being at least $C^2$-smooth, is unlikely to be identical to the real SDF. We can make a reasonable assumption that $f_\theta$ resembles the real SDF in the differentiable region that in the non-differentiable region. In light of this, it is reasonable to enforce the alignment between the gradients and the Hessian for points within a thin-shell space of appropriate width.

Based on the aforementioned analysis, we propose a loss function to promote the alignment of the Gradient-Hessian:

$$L_{\text{align}}(\boldsymbol{q}) = \|\mathbf{H}_{f_\theta} \boldsymbol{g}(\boldsymbol{q})\|_2^2 \tag{4}$$

with

$$\boldsymbol{g}(\boldsymbol{q}) = \frac{\nabla f_\theta(\boldsymbol{q})}{\|\nabla f_\theta(\boldsymbol{q})\|_2}. \tag{5}$$

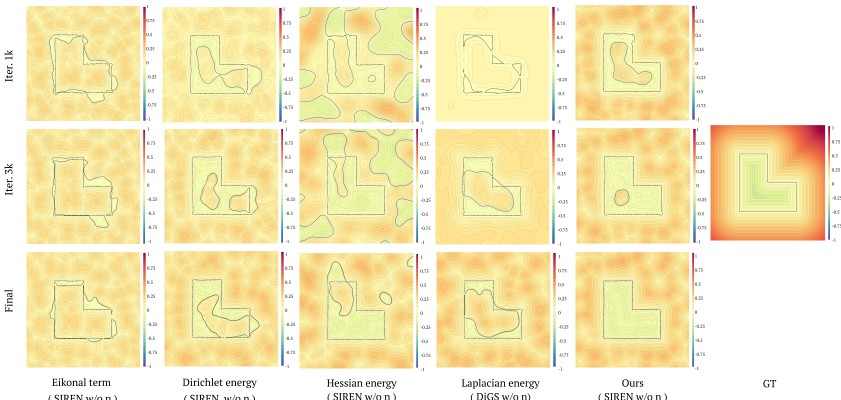

Figure 2: The level-sets, from left to right, show the distance fields learned by SIREN [40], SIREN with dirichlet energy [28], laplican energy [6], and hessian energy [54], DiGS [6] and ours with 100 points (black) as input. The black lines represent zero-isosurface. Our methods effectively suppress the ghost geometry with the concern of the gradient directions.

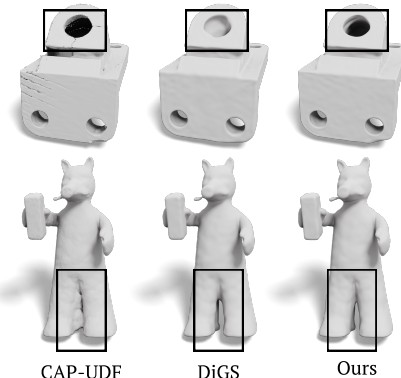

Figure 3: Visual comparisons on SRB [49].

| | Chamfer | | F-Score | |
| --- | --- | --- | --- | --- |
| | mean ↓ | std. ↓ | mean ↑ | std. ↓ |
| SPSR* [23] | 4.36 | 1.56 | 75.87 | 18.57 |
| SIREN [40] | 18.24 | 17.09 | 38.74 | 31.26 |
| SAP [38] | 6.19 | 1.75 | 57.21 | **11.66** |
| iPSR [20] | 4.54 | 1.78 | 75.07 | 19.18 |
| PCP [3] | 6.53 | 1.75 | 47.97 | 14.50 |
| CAP-UDF [55] | 4.54 | 1.82 | 74.75 | 18.84 |
| DiGS [6] | 4.16 | 1.44 | 76.69 | 18.15 |
| **Ours (SIREN)** | **3.86** | **1.10** | **78.80** | 16.01 |

Table 1: Quantitative comparison on SRB [49]. The methods marked with '*' require point normals.

**Difference from the Eikonal term.** By enforcing the Eikonal term, we can prevent the SDF from degenerating into a trivial field, such as $f_\theta = 0$ almost everywhere. But for a distance field rooted at any surface, $\|\nabla f_\theta\| = 1$ always holds except at the non-differentiable points. In practice, $\|\nabla f_\theta\| = 1$ is evaluated at discrete points, and the number of network parameters may significantly exceed the number of sample points. Consequently, specifying only $\|\nabla f_\theta\| = 1$ can result in ghost artifacts in the reconstructed surface, such as bulging effects, redundant parts, or excessive surface variations. For instance, consider the last row of Fig. 2. The reconstructed surfaces may differ significantly from the desired surface, even when $\|\nabla f_\theta\| = 1$ holds almost everywhere. This observation motivates us to seek more effective control over the gradient direction.

**Why not smoothness energy.** In previous literature, various forms of smoothness energy, such as Dirichlet Energy [28], Hessian energy [9, 54], and Laplacian energy [6], have been employed. Generally, the inclusion of smoothness terms can encourage smoothness in the SDF, reducing variations, and thereby mitigating the emergence of ghost geometries.

However, for these smoothing-based approaches, while they can reduce the occurrence of ghost geometry, they often sacrifice the ability to accurately represent geometric details. In contrast, our regularization term enforces the alignment between the gradient and the Hessian, which significantly differs from enforcing simple smoothness. As shown in Fig. 2, we created a 'L' shape with 100 boundary points, following the same experimental setup as DiGS [6], to illustrate the contrast between various approaches. Smoothing-based approaches tend not to reduce the smoothness energy to zero, even at termination. In contrast, our regularization term can approach zero more closely. Moreover, aligning the gradient and the Hessian follows an inherent property of the SDF, independent of the specific case. This sets our approach apart from smoothing-based methods that require case-by-case tuning of smoothing weights. As a result, our approach exhibits superior feature-preserving ability.

We conducted two ablation studies in Section 4.3, which clearly demonstrate that our alignment loss term not only more effectively suppresses the occurrence of ghost geometry but also yields more faithful reconstruction results than traditional smoothing-based approaches.

### 3.4 Loss function

Based on the aforementioned discussion, our neural function $f_\theta$ has to be at least $C^2$ continuous. In this paper, we consider two types of activation functions. The first activation function is Sine, which facilitates learning high-frequency information as pointed out in SIREN [40]. Obviously, Sine is $C^{\inf}$ continuous and satisfies our requirements. Additionally, Neural-Pull [5] leverages the SoftPlus activation function, which is a smooth version of ReLU and can also serve our purpose.

To this end, we establish the entire loss function by integrating the alignment loss into the existing loss configuration such that the network optimization can be formulated as

$$\arg\min_\theta \{E_{\text{old}}(f_\theta) + \alpha E_{\text{align}}(f_\theta)\},\tag{6}$$

| | Normal C. | | Chamfer | | F-Score | | IOU | |
|---|---|---|---|---|---|---|---|---|
| | mean↑ | std.↓ | mean↓ | std.↓ | mean↑ | std.↓ | mean↑ | std.↓ |
| SPSR* [23] | 90.58 | **3.30** | 4.66 | 4.64 | 75.28 | 25.76 | 90.58 | 6.77 |
| NSP* [50] | 90.74 | 5.48 | 8.85 | 6.96 | 52.42 | 28.55 | 79.08 | 12.41 |
| SAL [1] | 86.69 | 9.66 | 29.98 | 31.86 | 25.76 | 22.43 | 60.62 | 18.61 |
| IGR [19] | 80.85 | 11.88 | 62.54 | 48.44 | 26.28 | 35.91 | 33.33 | 24.51 |
| SIREN [40] | 83.79 | 10.20 | 34.19 | 46.77 | 32.34 | 30.13 | 41.14 | 15.82 |
| DiGS [6] | 95.82 | 4.44 | 4.59 | 4.94 | 78.87 | 27.34 | 89.54 | 8.83 |
| OSP [4] | 94.73 | 3.94 | 6.80 | 6.61 | 59.12 | 25.82 | 50.84 | 12.56 |
| iPSR [20] | 93.22 | 5.26 | 5.95 | 5.97 | 68.42 | 26.36 | 85.01 | 9.34 |
| PGR [27] | 91.90 | 4.93 | 7.34 | 4.81 | 51.44 | 23.12 | 78.34 | 11.17 |
| POCO⁺ [8] | 96.41 | 3.53 | 3.62 | 4.21 | 85.42 | 23.13 | **94.40** | **5.55** |
| **Ours (SIREN)** | **96.52** | 3.31 | **3.38** | **3.64** | **88.71** | **18.28** | 89.80 | 6.47 |

Table 2: Quantitative comparison of surface reconstruction under ShapeNet [13] with 3K points. The methods marked with '*' require normals, and the methods marked with '⁺' are supervision based.

where $E_{\text{align}}(f_\theta)$ is given by

$$\sum_{\boldsymbol{q} \in \mathcal{Q}} \beta_{\boldsymbol{q}} L_{\text{align}}(\boldsymbol{q}), \tag{7}$$

and $E_{\text{old}}(f_\theta)$ is the original loss function of existing works (the details will be discussed in the experiments). We leverage SIREN [40] without the normal version and Neural-Pull [5] in our experiment. Since the alignment effect between gradients and Hessian should be more emphasized near the underlying surface, we leverage an adaptive weighting scheme inspired by [15]:

$$\beta_{\boldsymbol{q}} = \exp\left(-\delta * |f_\theta(\boldsymbol{q})|\right), \tag{8}$$

we set $\delta = 10$ by default. At the same time, we set the default value of $\alpha$ to 6. More details can be checked in our supplementary material.

## 4 Experiments

**Metrics.** The indicators for comparison include normal consistency, chamfer distances, and F-Score, where normal consistency (%, abbreviated as 'Normal C.') reflects the degree to which the normals of the reconstructed surface agree with the normals of the ground-truth surface, chamfer distance (scaled by $10^3$, using $L_1$-norm) measures the fitting tightness between the two surfaces, and F-Score (%) indicates the harmonic mean of precision and recall (completeness). We set the default threshold of F-Score to 0.005. All meshes are uniformly scaled to $[-0.5, 0.5]$, and 100K points are sampled from each mesh for evaluation.

### 4.1 Optimization-based Surface Reconstruction

**Surface Reconstruction Benchmark (SRB).** The Surface Reconstruction Benchmark (SRB) [49] contains five shapes, each of which has challenging features, e.g., missing parts and rich details. The approaches for comparison include screened Poisson surface reconstruction (SPSR) [23], SIREN [40], Shape as points (SAP) [38], iPSR [20], Predictive Context Priors (PCP) [3], CAP-UDF [55] and DiGS [6]. Note that SPSR leverages normals given by the input scans. As shown in Tab. 1, our method outperforms the existing methods, as evidenced by superior results in both Chamfer distance and F-score metrics. In particular, the visual comparison presented in Fig. 3 demonstrates that our method is capable of accurately recovering the hole feature of the Anchor model, despite the absence of points on the inner wall. Furthermore, it successfully preserves the adjacent gaps of the Lord Quas model. In addition, we compared our approach with DiGS using SRB, following the evaluation settings of DiGS. Detailed performance statistics can be found in the supplementary material.

**ShapeNet.** The ShapeNet dataset [13] comprises a diverse collection of CAD models. We follow the splitting of [50] for the 13 categories of shapes with total of 260 shapes. Our comparison is performed using 3K points. The baseline methods include Screened Poisson Surface Reconstruction (SPSR) [23], NSP [50], SAL [1], IGR [19], SIREN [40], DiGS [6], OSP [4], iPSR [20] and PGR [27]. Note that SPSR and NSP require normal inputs. To ensure the validity of our experiments, we provide

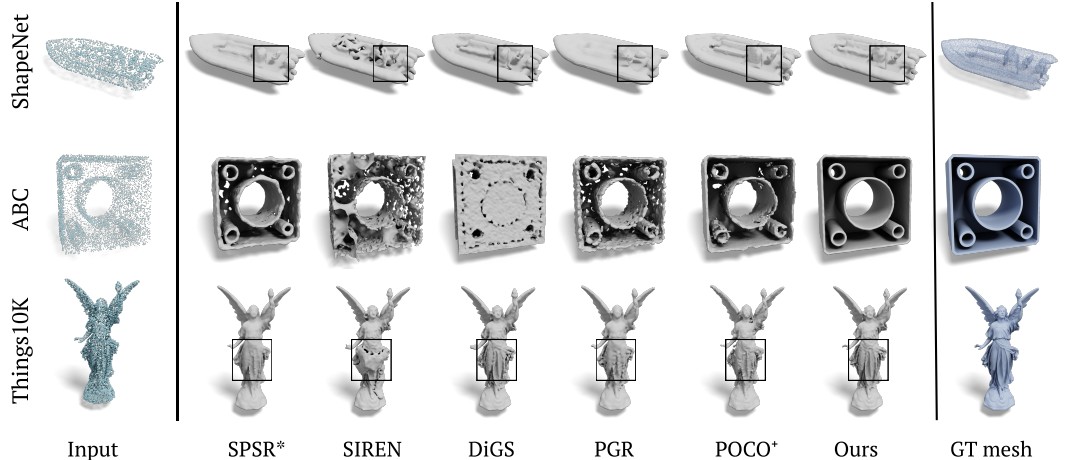

Figure 4: Comparison of our method to other methods under ShapeNet [13], ABC [24], and Thingi10K [57].

ground-truth normals to those that require normal information. Additionally, we include the learnable baselines Shape as Points (SAP) [38] and POCO [8] for comparison and retrain them from scratch on ShapeNet with 3K points. Based on the quantitative results presented in Tab. 2 and the visual comparison in Fig. 4. Our method effectively suppresses unnecessary surface variations and adapts the implicit representation to the inherent complexity encoded by the point cloud. Furthermore, our reconstruction accuracy is comparable to supervised methods. Specifically, our method outperforms POCO in Normal Correctness, Chamfer, and F-Score, but lags slightly behind POCO in terms of IOU.

**ABC and Thingi10K.** The ABC dataset [24] contains a variety of CAD meshes, while Thingi10K [57] comprises shapes with rich geometric details. We follow [16] to perform splitting for each dataset and randomly sample 10K points from each mesh. Each dataset contains 100 shapes. The baseline methods include Screened Poisson Surface Reconstruction (SPSR) [23], SAL [1], IGR [19], SIREN [40], Neural-Pull [5], Shape as Points (SAP) [38], DiGS [6], iPSR [20], and PGR [27]. Note that we found that the supervised version of SAP does not generalize well to shapes not present in the training set (ShapeNet) when using a global PointNet-based encoder. Therefore, we compare our approach against the unsupervised version of SAP. We also include the supervised method POCO [8] for comparison. To assess the generalization ability of supervised methods, we retrain them using 10K points on ShapeNet. The quantitative comparison results are presented in Tab. 3. Furthermore, the visual comparison on ABC [24] and Thingi10K [57] (see Fig. 4) demonstrates that our method can effectively recover CAD features such as small holes and thin plates and has ability to accurately recover high-fidelity geometric details.

**3D Scene.** Finally, we conducted tests on a 3D scene dataset [56] by sampling 20K points for each model. The quantitative comparison results are presented in Tab. 4 and visual comparison in Fig. 5. Our method, when combined with either SIREN [40] or Neural-Pull [5], yields significant improvements even with only 20K points for the scene-level reconstruction.

|  | Normal C. | | Chamfer | | F-Score | |
|---|---|---|---|---|---|---|
|  | mean↑ | std.↓ | mean↓ | std.↓ | mean↑ | std.↓ |
| SIREN [40] | 84.26 | 3.46 | 7.38 | 4.47 | 53.94 | 29.13 |
| **Ours (SIREN)** | **88.82** | **3.33** | **7.14** | **4.12** | **58.37** | 36.97 |
| Neural-Pull [5] | 82.05 | 5.34 | 33.96 | 17.14 | 20.99 | 25.46 |
| **Ours (Neural-Pull)** | **87.32** | **5.11** | **12.04** | **9.79** | **35.82** | 28.86 |

Table 4: Quantitative comparison on 3D Scene [56].

Notably, our method effectively captures both the details and overall shape of these challenging scenes.

## 4.2 Shape Space Learning

The D-Faust [7] dataset contains high-resolution raw scans (triangle soups) of 10 humans with various poses. We follow DualOctreeGNN [46] to perform splitting that 6K scans are used for training and

| | ABC | | | | | | Thingi10K | | | | | |
|---|---|---|---|---|---|---|---|---|---|---|---|---|
| | Normal C. | | Chamfer | | F-Score | | Normal C. | | Chamfer | | F-Score | |
| | mean↑ | std.↓ | mean↓ | std.↓ | mean↑ | std.↓ | mean↑ | std.↓ | mean↓ | std.↓ | mean↑ | std.↓ |
| SPSR* [23] | 95.16 | 4.48 | 4.39 | 3.05 | 74.54 | 26.65 | 97.15 | 2.95 | 3.93 | **1.79** | 77.03 | 23.71 |
| SAL [1] | 86.25 | 8.39 | 17.30 | 14.82 | 29.60 | 18.04 | 92.85 | 5.01 | 13.46 | 7.97 | 27.56 | 14.64 |
| IGR [19] | 82.14 | 16.12 | 36.51 | 40.68 | 43.47 | 40.06 | 90.20 | 10.61 | 27.80 | 34.8 | 54.28 | 39.99 |
| SIREN [40] | 82.26 | 9.24 | 17.56 | 15.25 | 30.95 | 22.23 | 88.30 | 6.53 | 17.69 | 13.47 | 26.20 | 19.74 |
| Neural-Pull [5] | 94.23 | 4.57 | 6.73 | 5.15 | 42.67 | **10.75** | 96.15 | 2.80 | 5.89 | 1.12 | 46.44 | **8.53** |
| SAP [38] | 81.59 | 10.61 | 15.18 | 16.60 | 45.88 | 33.67 | 92.60 | 7.03 | 10.61 | 13.84 | 53.32 | 31.91 |
| DiGS [6] | 94.48 | 6.12 | 6.91 | 6.94 | 66.22 | 32.01 | 97.25 | 3.30 | 5.36 | 5.59 | 74.45 | 27.11 |
| iPSR [20] | 93.15 | 7.47 | 4.84 | 4.06 | 71.59 | 24.96 | 96.46 | 3.57 | 4.41 | 2.94 | 74.88 | 22.72 |
| PGR [27] | 94.11 | 4.63 | 4.52 | 2.13 | 68.91 | 27.86 | 96.80 | 3.25 | 4.22 | 2.01 | 72.86 | 22.98 |
| POCO+ [8] | 92.90 | 7.00 | 6.05 | 6.80 | 68.29 | 26.05 | 95.16 | 5.00 | 5.61 | 9.42 | 73.92 | 25.79 |
| **Ours (SIREN)** | **96.50** | **3.95** | **3.72** | **2.48** | **83.50** | 19.28 | **97.92** | **2.90** | **3.34** | 3.14 | **89.68** | 17.11 |

Table 3: Quantitative comparison on ABC [24] and Thingi10K [57]. Each raw point cloud has 10K points.

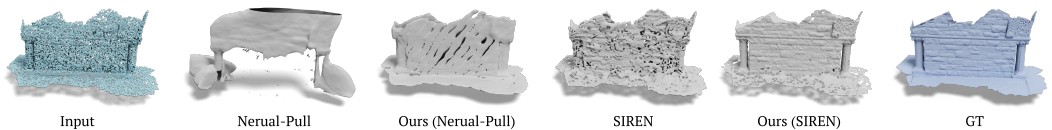

Figure 5: Visual comparison of our method to other methods under 3D Scene [56].

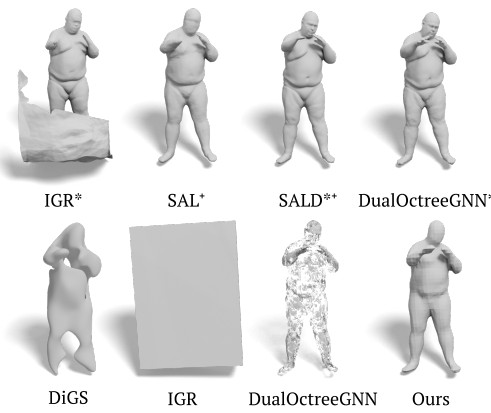

IGR*    SAL+    SALD*+    DualOctreeGNN*

DiGS    IGR    DualOctreeGNN    Ours

Figure 6: Visual comparison on DFAUST [7].

| | Normal C. | | Chamfer | | F-Score | |
|---|---|---|---|---|---|---|
| | mean↑ | std.↓ | mean↓ | std.↓ | mean↑ | std.↓ |
| IGR* [19] | 92.02 | 3.34 | 29.01 | 33.61 | 73.32 | 14.05 |
| DualOctreeGNN* [46] | **97.65** | **0.34** | **1.78** | 3.70 | **97.48** | 1.03 |
| SAL+ [1] | 96.77 | 0.81 | 2.82 | 4.67 | 91.35 | 9.15 |
| SALD*+ [2] | 97.04 | 0.92 | 3.06 | **1.32** | 88.56 | 12.73 |
| IGR [19] | 57.93 | 3.40 | 48.56 | 2.35 | 6.54 | **0.11** |
| DiGS [6] | 87.60 | 1.91 | 11.87 | 4.56 | 37.77 | 7.11 |
| DualOctreeGNN [46] | 92.42 | **0.45** | 3.02 | 2.38 | 85.77 | 3.50 |
| **Ours (SIREN)** | **95.43** | 0.41 | 2.95 | 2.21 | 88.75 | 2.51 |

Table 5: Quantitative comparison on DFAUST [7]. All methods marked with '*' means it leverages with normals in the training stage and marked with '+' means supervised.

2K scans for testing. In the training phase, we adopt the encoder following Convolutional Occupancy Networks [41]. Besides, we adopt the FiLM conditioning [11] that applies an affine transformation to the network's intermediate features as SIREN is weak in handling high-dimensional inputs [11, 33]. The baseline approaches include IGR [19], SAL [1], SALD [2], DualOctreeGNN [46] and DiGS [6]. Also, we further demonstrate the results of IGR, as well as the results of DualOctreeGNN trained without normals. As shown in Fig. 6, although IGR can generate well details, it yields spurious planes away from the input. SAL, supervised with unsigned distance, can only produce smooth results. Additionally, SALD, with the support of normal supervision, can generate more details, but its reconstruction accuracy is even worse than SAL since it suffers from a large systematic misalignment that does not respect input poses.

By contrast, DualOctreeGNN gets the most impressive results since the well-designed octree network can capture local prior for details. However, the performance of either IGR or DualOctreeGNN is compromised without normals.

| | Normal C. | | Chamfe | | F-Score | |
|---|---|---|---|---|---|---|
| | mean↑ | std.↓ | mean↓ | std.↓ | mean↑ | std.↓ |
| $\alpha = 0.06$ | 91.02 | 4.34 | 4.55 | 4.10 | 76.82 | 29.41 |
| $\alpha = 0.6$ | 92.65 | 3.95 | 4.17 | 4.20 | 80.44 | 23.46 |
| $\alpha = 6$ | **96.52** | **3.31** | **3.38** | 3.64 | **88.71** | **18.28** |
| $\alpha = 60$ | 93.04 | 1.03 | 2.19 | 0.73 | 82.91 | 24.73 |

Table 6: Effect of weights of $L_{\text{align}}$.

DiGS cannot yield reliable results though it is also based on SIREN. To summarize, our method can learn shape space without requiring input normals or additional supervision but still produces faithful shapes. It's important to note that the resulting surfaces of DualOctreeGNN for point clouds without input normals are not watertight, despite the small Chamfer distances.

## 4.3 Ablation Study

Our ablation studies were conducted on the ShapeNet dataset [13], which comprises 13 categories of shapes, with 20 shapes per category for a total of 260 shapes, each represented by 3K points. We use SIREN as the activation function for our baseline.

**Comparison with smoothness energies**    In Figure 7, we increased the weighting coefficient of the regularization term to $1\times10^3$, $1\times10^5$, and $1\times10^7$, respectively, for both our approach and those based on smoothing. It's evident that even with a large weight, our regularization term consistently produces high-fidelity surfaces, in contrast to the smoothing-based approaches, whereas their reconstruction results may become overly smooth or even fail (Note that double layers for Dirichlet energy).

A similar trend is observed in the 3D context. As illustrated in Figure 9, our method excels in reconstructing faithful and high-fidelity shapes from input data with 100K points, while the other approaches tend to produce over-smoothed results.

We also conducted an ablation study to compare the quantitative performance of our energy term with that of Dirichlet Energy [28] and Hessian Energy [6]. The results presented in Tab. 8 demonstrate that our energy term $E_{\text{align}}(f_\theta)$ significantly improves the performance. It's worth noting that the comparison between DiGS [6] (with a Laplacian energy) and ours has been made in previous subsections, which shows that ours has a superior performance. Therefore, in this subsection, we do not include Laplacian energy for comparison.

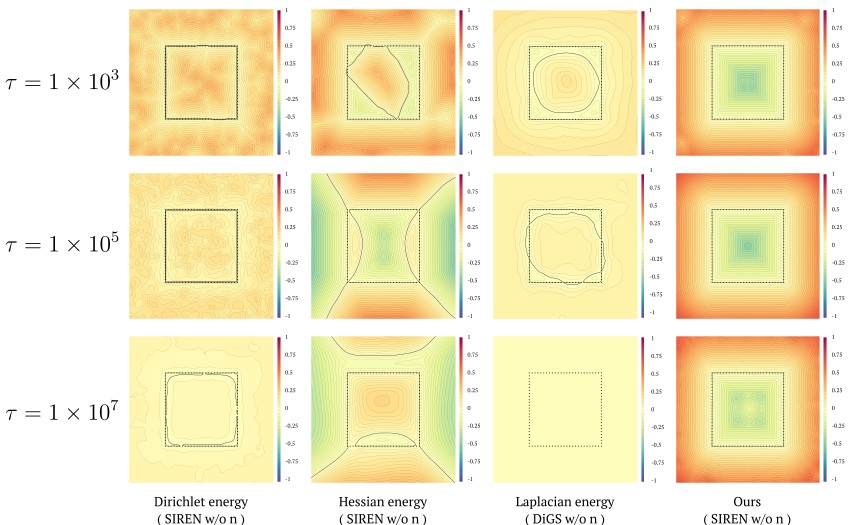

Figure 7: Ablation study on the weighting coefficient of the regularization term.

**Effect of weights of $L_{\text{align}}$.**    We conducted an ablation study to evaluate the effect of the weight $\alpha$ of $L_{\text{align}}$. By selecting different values, i.e., $\alpha = \{0.06, 0.6, 6, 60\}$, we keep the statistics in Tab. 6, which demonstrates that the best performance was achieved when the weight was set to 6.

**Effect of adaptive per points weights.** We demonstrate the effectiveness of the adaptive weight $\beta_q$ for each query in Tab. 7. We also compare the performance of different decay parameters $\delta$ used to compute $\beta_q$ in Eq. 8, and find that $\delta = 10$ yields the best results.

| | Normal C. | | Chamfer | | F-Score | |
|---|---|---|---|---|---|---|
| | mean↑ | std.↓ | mean↓ | std.↓ | mean↑ | std.↓ |
| $\delta = 0$ | 91.85 | 7.22 | 4.73 | 3.91 | 79.95 | 22.38 |
| $\delta = 1$ | 91.85 | 7.23 | 4.72 | 3.52 | 79.94 | 22.38 |
| $\delta = 10$ | **96.52** | **3.31** | **3.38** | **3.64** | **88.71** | **18.28** |
| $\delta = 100$ | 90.65 | 6.93 | 4.81 | 3.77 | 80.12 | 25.41 |

Table 7: Effect of adaptive per points weights.

**Comparison to LSA.** Recently, LSA [31] proposed enforcing alignment between the gradient of each level set and the gradient of the 0-level set to account for the direction of the gradient. However, it is important to note that the gradient of the 0-level set may not align with the true surface normals for its results. As such, using

|  | Normal C. | | Chamfer | | F-Score | |
|---|---|---|---|---|---|---|
|  | mean↑ | std.↓ | mean↓ | std.↓ | mean↑ | std.↓ |
| Dirichlet Energy | 92.91 | 7.48 | 4.30 | 3.75 | 77.67 | 23.83 |
| Hessian energy | 93.28 | 7.21 | 4.38 | 3.85 | 81.12 | 25.41 |
| **Ours** | **96.52** | **3.31** | **3.38** | **3.64** | **88.71** | **18.28** |

Table 8: Quantitative comparison to Smooth Energy.

the gradient of the 0-level set as a reference may not be particularly effective. We present results for SIREN [40] using both our loss and LSA under six shapes from the Stanford Scanning dataset with 2 million points same as the settings of LSA in its main paper. As shown in Fig. 8, our method effectively eliminates ghost geometry in empty areas due to incorrect gradient direction, while LSA does not. Tab. 9 presents the comparison statistics for three different approaches: 1) our methods combined with SIREN [40], 2) original SIREN and 3) LSA.

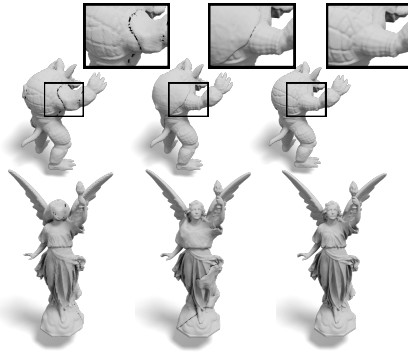

SIREN          LSA (SIREN)          Ours (SIREN)

|  | Normal C. | | Chamfer | | F-Score | |
|---|---|---|---|---|---|---|
|  | mean↑ | std.↓ | mean↓ | std.↓ | mean↑ | std.↓ |
| SIREN | 93.37 | 5.58 | 7.19 | 7.57 | 74.82 | 29.21 |
| LSA (SIREN) | 96.40 | 1.04 | 4.10 | 1.19 | 86.38 | 7.37 |
| **Ours (SIREN)** | **98.25** | **1.03** | **2.19** | **0.73** | **97.66** | **4.15** |

Table 9: Quantitative comparison to LSA [31].

Figure 8: Visual comparison with SIREN [40] and LSA [31] on Stanford Scanning dataset.

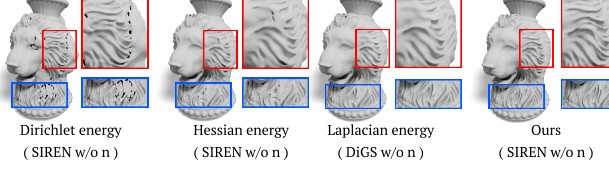

Dirichlet energy          Hessian energy          Laplacian energy          Ours
( SIREN w/o n )          ( SIREN w/o n )          ( DiGS w/o n )          ( SIREN w/o n )

Input     DiGS     Ours     Input     DiGS     Ours

Figure 9: Comparison with smoothness energies in 3D context.

Figure 10: Our method encountering challenges while handling sparse inputs.

# 5 Conclusion

In this paper, we propose a novel approach for surface reconstruction from unoriented point clouds by aligning the gradient and the Hessian of the Signed Distance Function (SDF). Unlike existing smoothness energy formulations that minimize SDF volatility, our approach offers more effective control over gradient direction, allowing the implicit function to adapt to the inherent complexity of the input point cloud. Comprehensive experimental results demonstrate that our approach effectively suppresses ghost geometry and recovers high-fidelity geometric details, surpassing state-of-the-art methods in terms of reconstruction quality.

Our current algorithm still has a few drawbacks. Firstly, it struggles with processing super sparse inputs, such as sketch point clouds or LiDAR data (as shown in Figure 10). The inherent challenge arises from completing extensive missing parts and closing the gaps between the input stripes. Secondly, it faces difficulties when dealing with large-scale scenes, like those from Matterport3D [12], mainly due to the neural networks' catastrophic forgetting issue. Representing large-scale scenes within a single network becomes exceedingly challenging. In our future work, we plan to address these challenges. To handle sparse inputs more effectively, we are considering implementing a supervision mechanism. Additionally, we aim to adopt a sliding window strategy, similar to approaches like DeepLS [10] and BlockNeRF [43], to mitigate the catastrophic forgetting issue.

# 6    Acknowledgement

The authors would like to thank the anonymous reviewers for their valuable comments and suggestions. This work is supported by the National Key R&D Program of China (2022YFB3303200), the National Natural Science Foundation of China (62002190, 62272277, 62072284), and the Key Research and Development Plan of Shandong Province of China (2020ZLYS01).

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
