# Supplementary Material: Aligning Gradient and Hessian for Neural Signed Distance Function

## A  Implementation Details

**Network design.**  As our method requires that the implicit function has at least second-order smoothness, we combine our method with SIREN [25] and Neural-Pull [5] that leverage the Softplus activation function. For SIREN-based MLP, we use a 4-layer SIREN-based MLP with 256 nodes in each layer. We use the Adam [16] optimizer with a learning rate of $5 \times 10^{-5}$, a batch size of 15K, and a total of 10K iterations. For combining our term with Neural-Pull [5], we use a 8-layers fully connected network architecture with 512 nodes in each layer. We use the Adam optimizer with a learning rate of 0.0001, a batch size of 10K, and 10K iterations. We also evaluate our method with multi-image input with NeuS [27]. We use the same architecture as Neural-Pull which is an 8-layer MLP with 256 nodes in each layer, and a skip connection is used to connect the input with the output of the fourth layer. We use the Adam [16] optimizer with a learning rate that is first linearly warmed up from 0 to 0.0005 in the first 5K iterations, then controlled by the cosine decay schedule to the minimum learning rate of 0.000025. We sample 512 rays per batch and train our model for 300k iterations (for the 'w/o mask' setting).

**Loss.**  As discussed in Section 3.4 of the main paper, our proposed loss term $E_{\mathrm{align}}\left(f_\theta\right)$ can be integrated with the existing methods by combining it with $E_{\mathrm{old}}\left(f_\theta\right)$. Due to our overfitting experiments were mainly done by combining with SIREN [25], we only modified the weights of $E_{\mathrm{old}}\left(f_\theta\right)$ in SIREN (without normal) from $W_{\mathrm{old}} = (3000, 100, 50)$ to $W_{\mathrm{new}} = (7000, 600, 50)$. When it comes to combining with other methods, i.e., Neural-Pull [5] and NeuS[27], we did not change the $W_{\mathrm{old}}$ in them. For $E_{\mathrm{align}}\left(f_\theta\right)$, according to ablation study results in the table in Section 4.3 of the main paper, we determined the initial weight $\alpha$ of this item to be 6. The annealing factor $\tau$ remains 6 during the first 30% iterations, then linearly decreases to 0.0001 during the 30% to 60% iterations, and finally decreases to 0 at the termination, *i.e.*, $\tau = (6, 0.3, 6, 0.6, 0.0001, 0)$. The annealing mechanism is followed by DiGS [6]. So we called the series of $\tau$ for $E_{\mathrm{align}}\left(f_\theta\right)$ in our paper as $D_{\mathrm{align}}$, and $D_{\mathrm{DiGS}}$ for DiGS. Here "D" stands for "decay".

**Sampling**  To evaluate our $E_{\mathrm{align}}$, we additional sample points near the surface for the methods with point clouds as input. Suppose that $p_i$ is a point in input $\boldsymbol{P}$, we define the Gaussian function rooted at $p_i$ and take the distance to its $k$-th nearest neighbor ($k = 50$) as the standard deviation. Then we sample points from each distribution. In the case of NeuS [27] with multi-image input, we adopt the original sampling points provided by NeuS on the ray for evaluation.

## B  Additional Experiments

### B.1  Ablation Studies

Our ablation studies are conducted over the ShapeNet [10] dataset where each shape is discretized into 3K points.

**Comparison to $W_{\mathbf{old}}$ and DiGS**  To further clarify that our method is independent of the $W_{\mathrm{new}}$ and $D_{\mathrm{align}}$ assigned. We designed our ablation study with DiGS. DiGS [6] is also an unoriented point

37th Conference on Neural Information Processing Systems (NeurIPS 2023).

cloud reconstruction method based on SIREN [25]. Unlike us, DiGS takes the $W_{\text{old}}$ and has its own weights $D_{\text{DiGS}} = \{100, 0.2, 100, 0.4, 0, 0\}$ of its loss term $E_{\text{DiGS}}(f_\theta)$. Therefore, by incorporating DiGS, we can effectively compare $W_{\text{old}}$ and $W_{\text{new}}$ as well as assess the differences between $D_{\text{DiGS}}$ and $D_{\text{align}}$, and it is a straightforward and intuitive way to compare our methods with DiGS. We conduct comprehensive experiments with DiGS [6] and show the statistics in Tab. 1, which shows that when we apply $W_{\text{new}}$ or $D_{\text{align}}$ to DiGS, the results deteriorated. Similarly, when $W_{\text{old}}$ or $D_{\text{DiGS}}$ are applied to our method, they do not yield effective results.

| | $W_{\text{old}}$ | $W_{\text{new}}$ | $D_{\text{DiGS}}$ | $D_{\text{align}}$ | Normal C. ↑ | | Chamfer ↓ | | F-Score ↑ | |
| --- | --- | --- | --- | --- | --- | --- | --- | --- | --- | --- |
| | | | | | mean | std. | mean | std. | mean | std. |
| DiGS [6] | ✓ | | ✓ | | 95.82 | 4.44 | 4.59 | 4.94 | 78.87 | 27.34 |
| | ✓ | | | ✓ | 92.19 | 6.37 | 6.42 | 5.75 | 70.11 | 30.67 |
| | | ✓ | ✓ | | 91.92 | 7.08 | 4.79 | 3.54 | 77.73 | 21.96 |
| | | ✓ | | ✓ | 91.02 | 7.08 | 4.83 | 3.44 | 73.77 | 25.32 |
| **Ours (SIREN)** | ✓ | | ✓ | | 90.92 | 6.99 | 6.94 | 6.51 | 66.50 | 30.73 |
| | ✓ | | | ✓ | 91.55 | 6.90 | 7.65 | 7.14 | 64.99 | 33.62 |
| | | ✓ | ✓ | | 93.06 | 6.52 | 5.07 | 5.33 | 80.17 | 25.17 |
| | | ✓ | | ✓ | **96.52** | **3.31** | **3.38** | **3.64** | **88.71** | **18.28** |

Table 1: Quantitative comparison with DiGS [6] and ablation study of loss weights.

| | Mean | |
| --- | --- | --- |
| | GT | |
| | $d_C$ | $d_H$ |
| Bbox-sampling | 0.32 | 4.51 |
| k=1 | 0.32 | 4.90 |
| k=25 | 0.28 | 5.87 |
| k=50 (ours) | **0.19** | **2.98** |
| k=75 | 0.26 | 4.31 |
| k=100 | 0.28 | 5.34 |

Table 2: Effects of different sampling strategies under SRB.

**Effect of sampling strategy**    Instead of uniformly sampling within the bounding box, we adopt a strategy similar to IGR [11] and NeuralPull [5], which involves sampling points around the input point set.

We have two primary reasons for this choice. Firstly, our primary interest lies in the 0-isosurface rather than other level sets. While the inferred distance field might exhibit slight differences from actual distances, this discrepancy doesn't negatively impact the 0-isosurface. Secondly, we intend to enforce the alignment between the gradient and the Hessian only within a thin shell encompassing the true surface. This alignment property ceases to hold outside the thin shell, as the SDF may become non-differentiable.

To evaluate various sampling strategies, we conducted comparisons under SRB [29]. The comparative statistics in Table 2 indicate that sampling around the surface is a better strategy. Specifically, $k = 50$ is the recommended choice.

**Comparison to initialization methods**    We investigate the effect of the multi-frequency geometric initialization (MFGI), the results in Tab. 3 show it produces a little better performance. However, it can not handle concave parts of shape for ours and DiGS with MFGI, see Fig. 1. Further, unlike our approach, DiGS cannot consistently yield better results if switching to the SIREN initiation.

## B.2    Run-time Performance

The second-order optimization increases the overhead of the back-propagation. We include IGR [11], SIREN [25] and DiGS [6] for comparison. We set the batch size to 15K for all the methods and utilized the network with four hidden layers, 256 units for each layer for the SIREN-based methods,

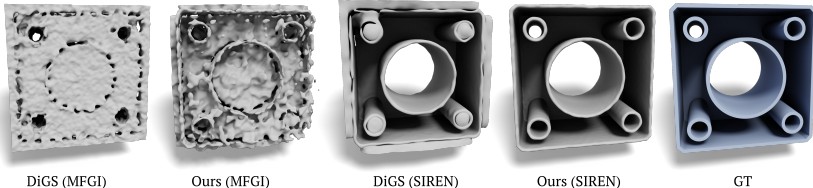

| DiGS (MFGI) | Ours (MFGI) | DiGS (SIREN) | Ours (SIREN) | GT |

Figure 1: Visual comparison of our method to DiGS under different initialization methods.

|  | Normal C. ↑ | | Chamfer ↓ | | F-Score ↑ | |
|---|---|---|---|---|---|---|
|  | mean | std. | mean | std. | mean | std. |
| Ours (SIREN init.) | 96.52 | 3.31 | 3.38 | 3.64 | 88.71 | 18.28 |
| Ours (MFGI init.) | **96.53** | **3.29** | **3.14** | **2.69** | **89.64** | **17.59** |

Table 3: Ablation study about initialization methods with MFGI.

which are the default setting for our method. Tab. 4 reports the timing cost spent in a single iteration. Roughly speaking, the timing costs of DiGS and ours are higher than SIREN since DiGS and ours need a second-order optimization. However, ours is more computationally efficient than IGR.

|  | IGR | SIREN | DiGS | Ours |
|---|---|---|---|---|
| # parameters | 1.86M | 264.4K | 264.4K | 264.4K |
| time [ms] | 50.73 | 11.52 | 36.28 | 40.10 |

Table 4: Timing costs per iteration. The comparison is made among IGR [11], SIREN [25], and DiGS [6] without the supervision of normals. Timing statistics are reported in milliseconds (ms).

## C  Evaluation Metrics

To compare the performance of different reconstruction methods, we use the same evaluation metrics as ConvONet [26], *i.e.*, Chamfer distances, F-Score, and Normal consistency. We denote $M_g$ and $M_p$ as the ground-truth mesh (or point cloud) and the mesh of the predicted result, respectively. Let $P_1$ and $P_2$ be the randomly sampled points on the ground-truth mesh (or point cloud) and the predicted mesh.

**Chamfer Distance**  The Chamfer distance between two point clouds $P_1$, $P_2$ is defined as follows:

$$\text{Chamfer}\,(P_1, P_2) = \frac{1}{2|P1|} \sum_{p_1 \in P_1} \min_{p_2 \in P_2} d(p_1, p_2) \\ + \frac{1}{2|P2|} \sum_{p_2 \in P_2} \min_{p_1 \in P_1} d(p_1, p_2), \tag{1}$$

where $d(p_1, p_2)$ is the straight-line distance between points $p1$, $p2$. We use the $L_1$ norm following ConvONet [26].

**F-Score**  The F-Score between the two point clouds $P_1$ and $P_2$ at a given threshold $t$ is given by:

$$\text{F-Score}\,(t, P_1, P_2) = \frac{2\,\text{Recall}\,\text{Precision}}{\text{Recall}\,+\,\text{Precision}}, \tag{2}$$

where

$$\text{Recall}\,(t, P_1, P_2) = \left| \left\{ p_1 \in P_1, \text{s.t.} \min_{p_2 \in P_2} d\,(p_1, p_2) < t \right\} \right|$$

$$\text{Precision}\,(t, P_1, P_2) = \left| \left\{ p_2 \in P_2, \text{s.t.} \min_{p_1 \in P_1} d\,(p_2, p_1) < t \right\} \right| \tag{3}$$

**Normal consistency**  The normal consistency between two point clouds $P_1$, $P_2$ is defined as follows:

$$\text{NormalC.}\,(P_1, P_2) = \frac{1}{2|P1|} \sum_{p_1 \in P_1} n_{p_1} \cdot n_{\text{closest}\,(p_1, P_2)}$$
$$+ \frac{1}{2|P2|} \sum_{p_2 \in P_2} n_{p_2} \cdot n_{\text{closest}\,(p_2, P_1)}, \tag{4}$$

where

$$\text{closest}(p, P) = \arg\min_{p' \in P} d\,(p, p') \tag{5}$$

# D  Experimental Setting for Separate Dataset

## D.1  SRB

We evaluated the baselines using their official source code. All methods utilized $256^3$ grids (SPSR [15] and iPSR [12] used octrees of depth 8) to extract the final mesh. We trained DiGS and SIREN with four hidden layers, each containing 256 units, and the total number of iterations was set to 10K, the same as in our method. Other parameters for each method were used with their default settings.

In Table 5, we present the relevant comparison statistics on the Surface Reconstruction Benchmark [29]. Our method achieves the highest scores for all shapes except for a slight gap compared to Daratech and DC. Visual comparisons are shown in Fig. 2.

We also conducted a comparison between our method and DiGS, following DiGS' evaluation setting. Table 6 illustrates that our method outperforms DiGS in terms of Hausdorff distance. We explain more about DiGS' evaluation setting. DiGS operates at a resolution of $512^3$ and uses Chamfer distance and Hausdorff distance at the original scale. However, DiGS does not specify the number of evaluation points. On the other hand, the Shape as Points version operates at a resolution of $256^3$ and employs Chamfer distance, F-Score, and Normal Consistency. To ensure a fair comparison, we adopted DiGS' metrics but sampled 100K evaluation points, following the settings of Shape as Points in this paper.

| | Mean | | Anchor | | Daratech | | DC | | Gargoyle | | Lord Quas | |
|---|---|---|---|---|---|---|---|---|---|---|---|---|
| | Chamfer ↓ | F-Score ↑ | Chamfer ↓ | F-Score ↑ | Chamfer ↓ | F-Score ↑ | Chamfer ↓ | F-Score ↑ | Chamfer ↓ | F-Score ↑ | Chamfer ↓ | F-Score ↑ |
| SPSR* | 4.36 | 75.87 | 6.93 | 46.14 | 4.20 | 83.22 | 3.40 | 85.89 | 4.37 | 70.65 | 2.85 | 93.60 |
| SIREN | 18.24 | 38.74 | 38.31 | 5.05 | 6.19 | 52.30 | 46.24 | 75.47 | 35.50 | 7.25 | 6.53 | 54.58 |
| SAP | 6.19 | 57.21 | 8.33 | 46.73 | 7.76 | 48.42 | 5.11 | 60.34 | 4.27 | 75.66 | 5.54 | 54.61 |
| iPSR | 4.54 | 75.07 | 7.53 | 44.29 | 4.20 | 83.51 | 3.52 | 84.36 | 4.49 | 69.87 | 2.91 | 93.53 |
| PCP | 6.53 | 47.97 | 9.04 | 37.63 | 7.23 | 36.08 | 5.82 | 45.09 | 6.17 | 49.71 | 4.30 | 72.09 |
| CAP-UDF | 4.54 | 74.75 | 7.68 | 43.92 | 3.96 | 82.78 | 3.61 | 84.03 | 4.40 | 70.82 | 3.06 | 92.19 |
| DiGS | 4.16 | 76.69 | 6.63 | 46.52 | 3.62 | **85.54** | **3.32** | **86.11** | 4.19 | 73.34 | 3.04 | 91.86 |
| **Ours(SIREN)** | **3.86** | **78.80** | **5.63** | **52.50** | **3.44** | 84.95 | 3.45 | 85.12 | **4.06** | **76.76** | **2.73** | **94.66** |

Table 5: Comparison on Surface Reconstruction Benchmark [29].

| | Mean | | Anchor | | | | Daratech | | | | DC | | | | Gargoyle | | | | Lord Quas | | | |
|---|---|---|---|---|---|---|---|---|---|---|---|---|---|---|---|---|---|---|---|---|---|---|
| | GT | | GT | | Scans | | GT | | Scans | | GT | | Scans | | GT | | Scans | | GT | | Scans | |
| | $d_C$ | $d_H$ | $d_C$ | $d_H$ | $d_{\bar{C}}$ | $d_{\bar{H}}$ | $d_C$ | $d_H$ | $d_{\bar{C}}$ | $d_{\bar{H}}$ | $d_C$ | $d_H$ | $d_{\bar{C}}$ | $d_{\bar{H}}$ | $d_C$ | $d_H$ | $d_{\bar{C}}$ | $d_{\bar{H}}$ | $d_C$ | $d_H$ | $d_{\bar{C}}$ | $d_{\bar{H}}$ |
| DiGS | **0.19** | 3.52 | 0.29 | 7.19 | 0.11 | 1.17 | **0.20** | 3.72 | 0.09 | 1.80 | 0.15 | **1.70** | 0.07 | 2.75 | **0.17** | **4.10** | 0.09 | 0.92 | **0.12** | **0.91** | 0.06 | 0.70 |
| **Ours(SIREN)** | **0.19** | **2.98** | **0.28** | **4.79** | 0.24 | 1.78 | **0.20** | **2.52** | 0.13 | 1.84 | **0.14** | 1.88 | 0.10 | 2.77 | 0.19 | 4.56 | 0.15 | 1.82 | 0.14 | 1.13 | 0.09 | 0.95 |

Table 6: Comparison on Surface Reconstruction Benchmark [29] using the evaluation settings of DiGS[6].

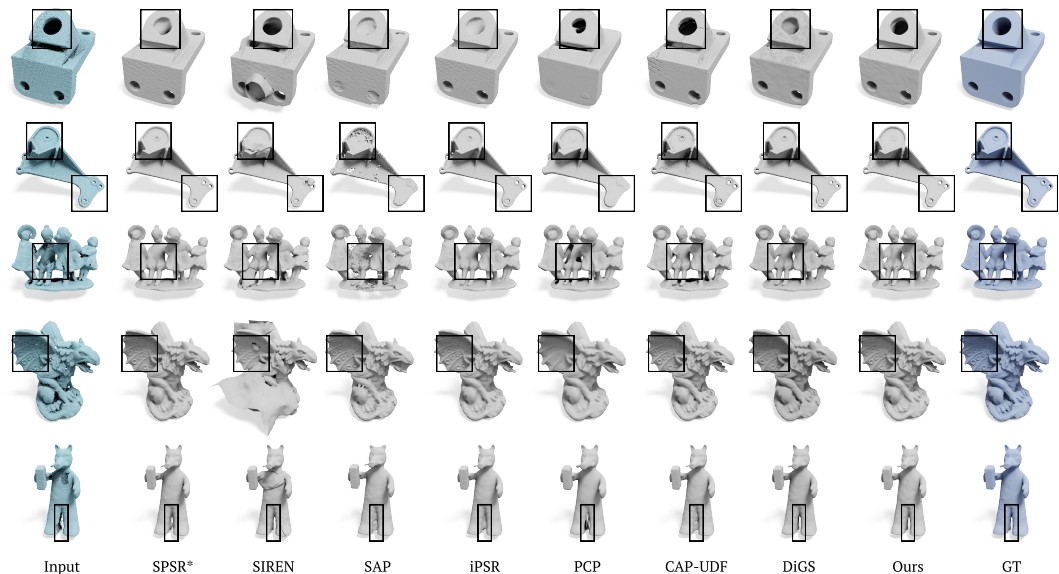

| Input | SPSR* | SIREN | SAP | iPSR | PCP | CAP-UDF | DiGS | Ours | GT |

Figure 2: Visual comparison of our method to other methods under SRB [29].

## D.2 ABC and Thingi10K

We report the results of baselines using their source code. All methods leverage $256^3$ grids, and SPSR [15], iPSR [12] and PGR [19] use the depth 8 during the mesh extraction phase. For SAL [1] and IGR [11], we trained them with 20K iterations and 15K iterations, respectively. We conduct 10K iterations for DiGS and SIREN, the same as ours, where the SIREN network has four hidden layers, each containing 256 neurons. For PGR, we use the officially recommended parameters for the 10K-point input (alpha: 1.2, wk: 16). For the supervision methods POCO [8] and Neural Galerkin [13] (without normals), we retrained them with 10K points under ShapeNet [10] to validate their generalization ability. Other parameters remain the same with the default settings.

We show the visual comparison of different approaches on ABC [17] with 10K points in Fig. 3 and Fig. 4. The comparison shows that Our method is better at recovering thin geometry features and can achieve a good trade-off between smoothness and feature preservation.

## D.3 ShapeNet

We report all baselines using their code. All methods leverage $256^3$ grids (SPSR [15], iPSR [12], and PGR [19] use the octree of depth 8) to extract the final mesh. We trained DiGS and SIREN with four hidden layers, each layer containing 256 units. The total number of iterations is set to 10K. We conduct 10K iterations for DiGS and SIREN, the same as ours, and train SAL and IGR within 20K iterations and 15K iterations, respectively. For NSP [30], we follow the parameters used in its main paper (1024 input points with 1024 Nyström samples and no regularization) and set the Nyström samples to 3000 for 3K input points, respectively, without regularization.For PGR, we use the officially recommended parameters for sparse inputs (alpha: 2, wmin: 0.04). For the supervision methods POCO [8], we retrain them with 3K points under ShapeNet [10], respectively. More parameters of each method follow the default setting. We give the comparison statistics under the settings of in Tab. 9. The visual comparison is given Fig. 5. Both qualitative and quantitative comparisons show that our method can faithfully recover fine geometric details and thin structures, outperforming the other methods.

## D.4 DFAUST

Shape space learning requires training a single model to learn to represent multiple shapes from a class of related shapes, which is more challenging than the single overfitting shape. For the encoder, we adopt the encoder from Convolutional Occupancy Network [26]. Specifically, we project the

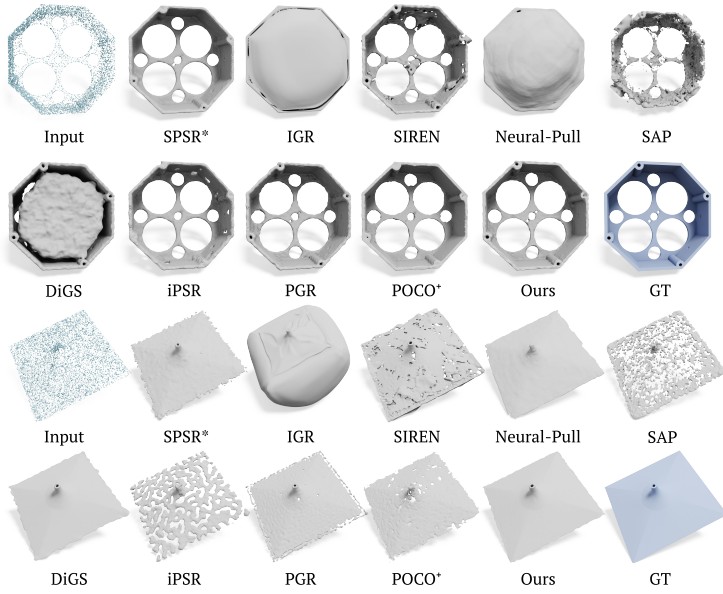

Figure 3: Visual comparison of our method to other methods under ABC [17].

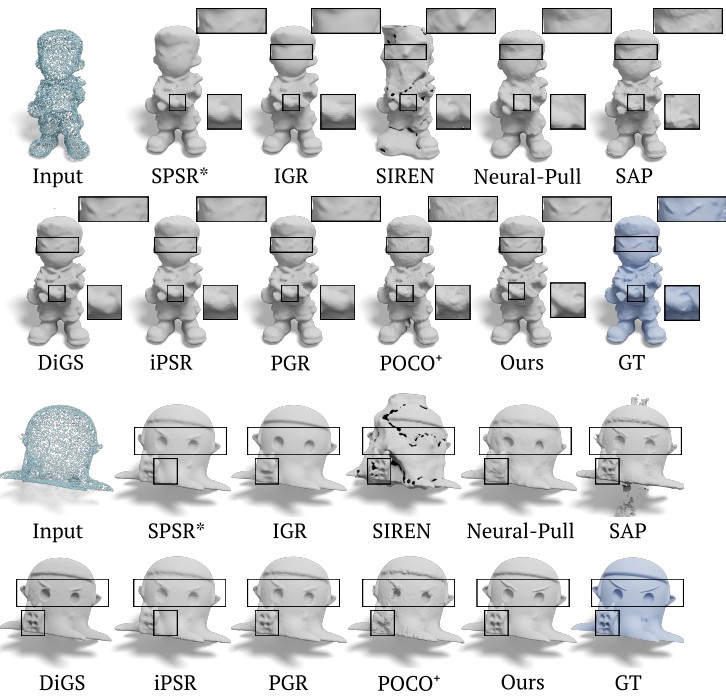

Figure 4: Visual comparison of our method to other methods under Thingsi10K [32].

sparse on-surface point features obtained using a modified PointNet [23] onto a regular 3D grid, then use a convolutional module to propagate sparse on-surface point features to the off-surface area, and finally obtain the query feature using bilinear interpolation. For the decoder, we use the SIREN network has three hidden layers. Further, we adopt the FiLM conditioning [9] that applies an affine transformation to the network's intermediate features as SIREN is weak in handling high-dimensional inputs [9, 22]. We train our models for 200 epochs using AMSGrad optimizer [24] with an initial learning rate of 0.0001 and decay to 0.000001 using cosine annealing [20]. We divided the training set into mini-batches: a batch contains 32 different shapes (accumulate batches), where each shape is

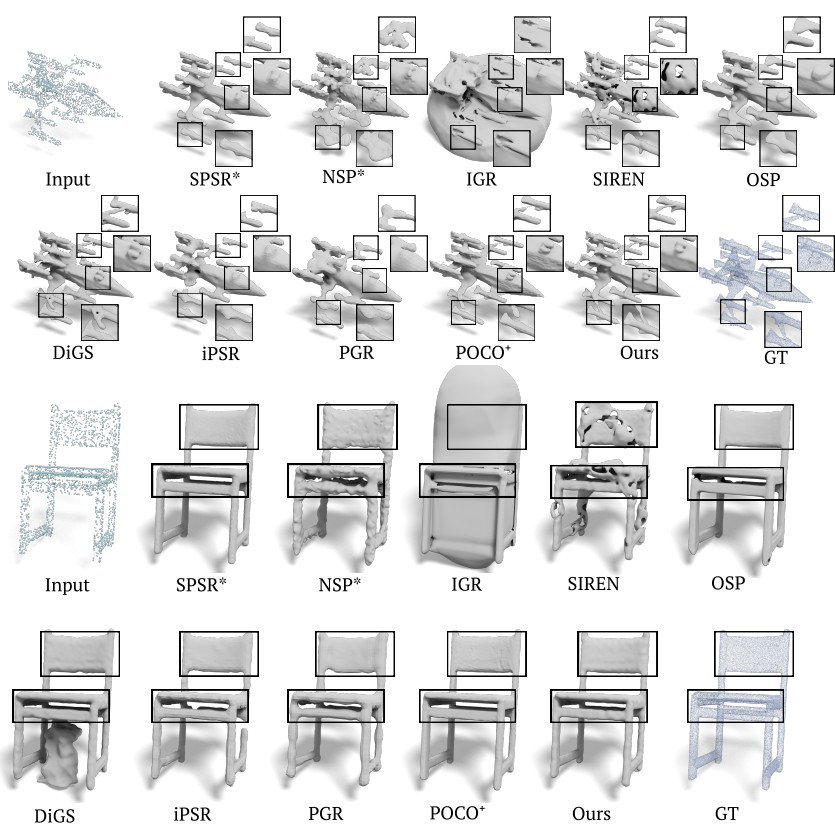

| | | | | | |
|---|---|---|---|---|---|
| Input | SPSR* | NSP* | IGR | SIREN | OSP |
| DiGS | iPSR | PGR | POCO+ | Ours | GT |

| | | | | | |
|---|---|---|---|---|---|
| Input | SPSR* | NSP* | IGR | SIREN | OSP |
| DiGS | iPSR | PGR | POCO+ | Ours | GT |

Figure 5: Visual comparison of our method to other methods under ShapeNet [10].

randomly sampled to produce 10K points. The experiments are conducted with 8 RTX 3090 graphics cards.

For baselines, we use the pre-trained model of IGR [11], SAL [1], SALD [2], DualOctreeGNN [28], and DiGS [6], we also retrain the IGR and DualOctreeGNN for the version without normals supervision. Therefore, this baseline is omitted. The visual comparison of different approaches on the DFAUST [7] dataset is available in Fig. 6.

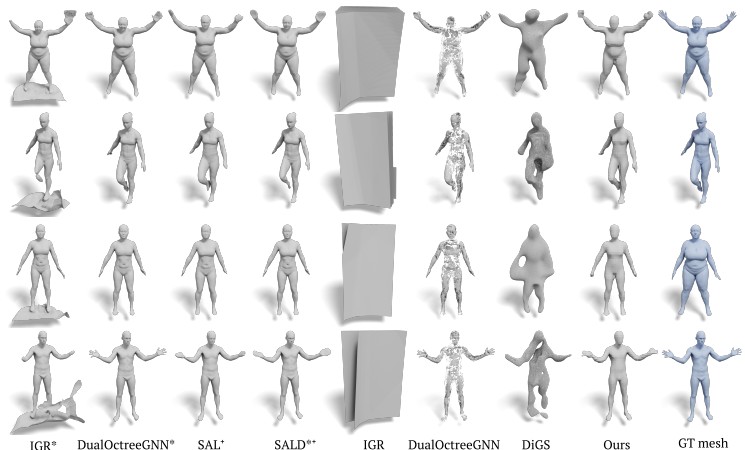

IGR*    DualOctreeGNN*    SAL+    SALD**    IGR    DualOctreeGNN    DiGS    Ours    GT mesh

Figure 6: Visual comparison of shape space learning on DFAUST [7].

## D.5 DTU

Here we also employ our loss to NeuS [27] with multi-image input. NeuS also leverage SoftPlus activation same as NeuralPull [5] and IGR [11] to enable second-order optimization. We sample 512 rays per batch and train our model for 300k iterations as NeuS's training settings ('w/o mask'). Further, we compared our method with LSA combined with NeuS. Fig. 7 and Tab. 7 both shows that our methods consistently improve the accuracy for multi-image input. Note that our method achieves comparable performance to LSA with multi-image input, but it outperforms LSA when it comes to point cloud reconstruction, as shown in main paper.

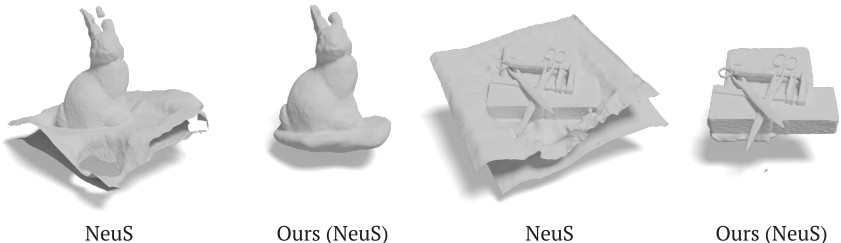

| NeuS | Ours (NeuS) | NeuS | Ours (NeuS) |

Figure 7: visual comparison to NeuS [27] on DTU dataset [14].

| | Mean Chamfer ↓ | scan_24 Chamfer ↓ | scan_37 Chamfer ↓ | scan_40 Chamfer ↓ | scan_55 Chamfer ↓ | scan_63 Chamfer ↓ | scan_65 Chamfer ↓ | scan_69 Chamfer ↓ | scan_83 Chamfer ↓ | scan_97 Chamfer ↓ | scan_105 Chamfer ↓ | scan_106 Chamfer ↓ | scan_110 Chamfer ↓ | scan_114 Chamfer ↓ | scan_118 Chamfer ↓ | scan_122 Chamfer ↓ |
|---|---|---|---|---|---|---|---|---|---|---|---|---|---|---|---|---|
| NeuS [27] | 0.84 | 1.00 | 1.37 | 0.93 | 0.43 | 1.10 | 0.65 | **0.57** | 1.48 | 1.09 | **0.83** | 0.52 | 1.20 | 0.35 | 0.49 | 0.54 |
| LSA (NeuS) [21] | **0.78** | 0.88 | 0.90 | **0.80** | **0.41** | 1.13 | **0.63** | 0.58 | 1.37 | 1.16 | **0.83** | **0.51** | 1.26 | **0.33** | 0.48 | **0.52** |
| **Ours (NeuS)** | **0.78** | **0.83** | **0.88** | 0.92 | 0.58 | **0.96** | 0.82 | 0.98 | **0.94** | **0.89** | 0.89 | 0.61 | **0.91** | 0.44 | **0.43** | 0.58 |

Table 7: Comparison on DTU dataset [14].

## D.6 Large Scans

To evelute the geneiablity of supervised methods, we test the ability to handle huge-size point clouds using three shapes from ThreedScans [18]. We randomly sample about 300K points from each shape. We include SIREN [25] with ground-truth oriented normals, PCP [3], DiGS [6] and the learnable method of POCO [8] for comparison. All methods we used leverage $512^3$ grids to extract the mesh. We conduct 50K iterations for DiGS[6] and SIREN[25], where the SIREN network has four hidden layers, each containing 256 neurons, the same as ours, For the supervision method POCO [8], we retrained them with 10K points under ShapeNet [10] to validate its generality. Other parameters remain the same with the default settings.

The quantitative comparison statistics are reported in Tab. 8 and the visual comparison is available in Fig. 8. PCP and DiGS tend to produce smooth results without geometry details, where the smoothing energy of DiGS weakens the geometric details. POCO is supervision based and thus weak in the generalization ability with different point clouds resolution not available in the trainset even with Test Time Augmentation. It can be observed that, apart from SIREN, our method can better reconstruct richer and more detailed features.

| | | Mean | | | Std. | | | Eagle | | | Dragon | | | Hosmer | |
|---|---|---|---|---|---|---|---|---|---|---|---|---|---|---|---|
| | Normal C. ↑ | Chamfer ↓ | F-Score ↑ | Normal C. | Chamfer | F-Score | Normal C. ↑ | Chamfer ↓ | F-Score ↑ | Normal C. ↑ | Chamfer ↓ | F-Score ↑ | Normal C. ↑ | Chamfer ↓ | F-Score ↑ |
| SIREN* [25] | **98.38** | 0.96 | 63.61 | **0.30** | **0.17** | 19.62 | **98.74** | 1.16 | 42.55 | **98.21** | 0.84 | 81.40 | **98.21** | 0.87 | 66.89 |
| PCP [3] | 94.32 | 4.44 | 11.83 | 2.58 | 1.41 | 9.41 | 97.27 | 3.01 | 12.00 | 93.16 | 5.83 | 2.33 | 92.51 | 4.47 | 21.15 |
| DiGS [6] | 97.41 | **0.92** | 63.78 | 0.62 | 0.22 | 21.91 | 98.08 | **1.10** | 42.24 | 97.31 | 0.67 | 86.06 | 96.84 | 0.98 | 63.04 |
| POCO+ [8] | 85.97 | 3.02 | 31.01 | 9.10 | 2.09 | 20.67 | 95.89 | 1.43 | 46.66 | 77.99 | 5.39 | 7.57 | 84.03 | 2.24 | 38.79 |
| **Ours (SIREN)** | 97.44 | 0.99 | **98.91** | 0.59 | 0.41 | **1.69** | 98.10 | 1.45 | **96.95** | 97.11 | **0.70** | **99.93** | 97.04 | **0.81** | **99.85** |

Table 8: Quantitative comparison on the shapes from [18].

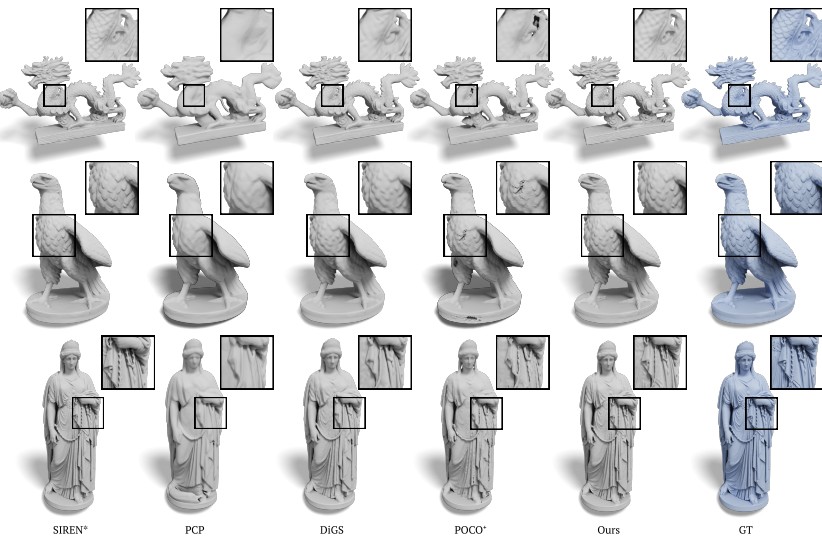

Figure 8: Visual comparison of different approaches on ThreedScans [18] with 10K points. Our method is comparable to the "with normals" version of SIREN.

# E  More Results

## E.1  3D Scene

We follow the default training setting on 3D Scene[31] dataset. We provide additional visual results in Fig. 9. Our method, when combined with either SIREN [25] or Neural-Pull [5], yields significant improvements even with only 20K points for the scene-level reconstruction.

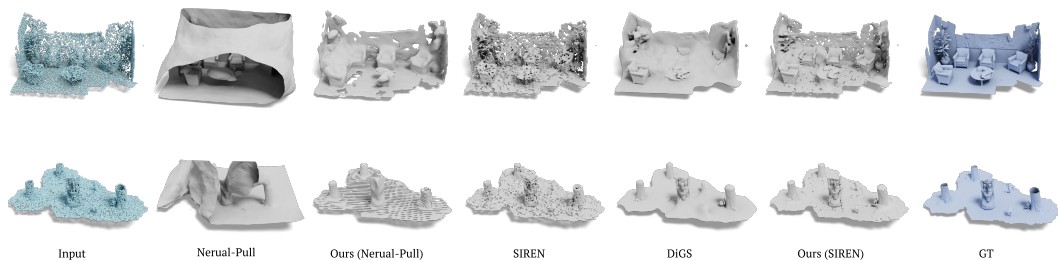

Figure 9: visual comparison on 3D Scene[31].

## E.2  Noisy Input

To test whether our method can handle noisy point clouds, we conducted experiments with SIREN [25] to evaluate its performance with noisy input.

We add Gaussian noise to the three shapes with different standard variations $\gamma = \{0.005, 0.01, 0.02\}$. At the same time, we adjust our weights $\alpha$ of term $E_{\text{align}}$ to $\alpha = \{0.06, 0.6, 6, 60\}$. We also modified $D_{\text{align}}$, to be specific, we either kept the original decay strategy or avoided decay by maintaining a fixed $\tau$. Each of the sampled point clouds consists of 20,000 points. The schedule followed our former training setting on SIREN. The visual results can be seen in Fig. 10. From the results, it can be observed that larger weights can yield smoother results and our method is robust to small noise with default configurations.

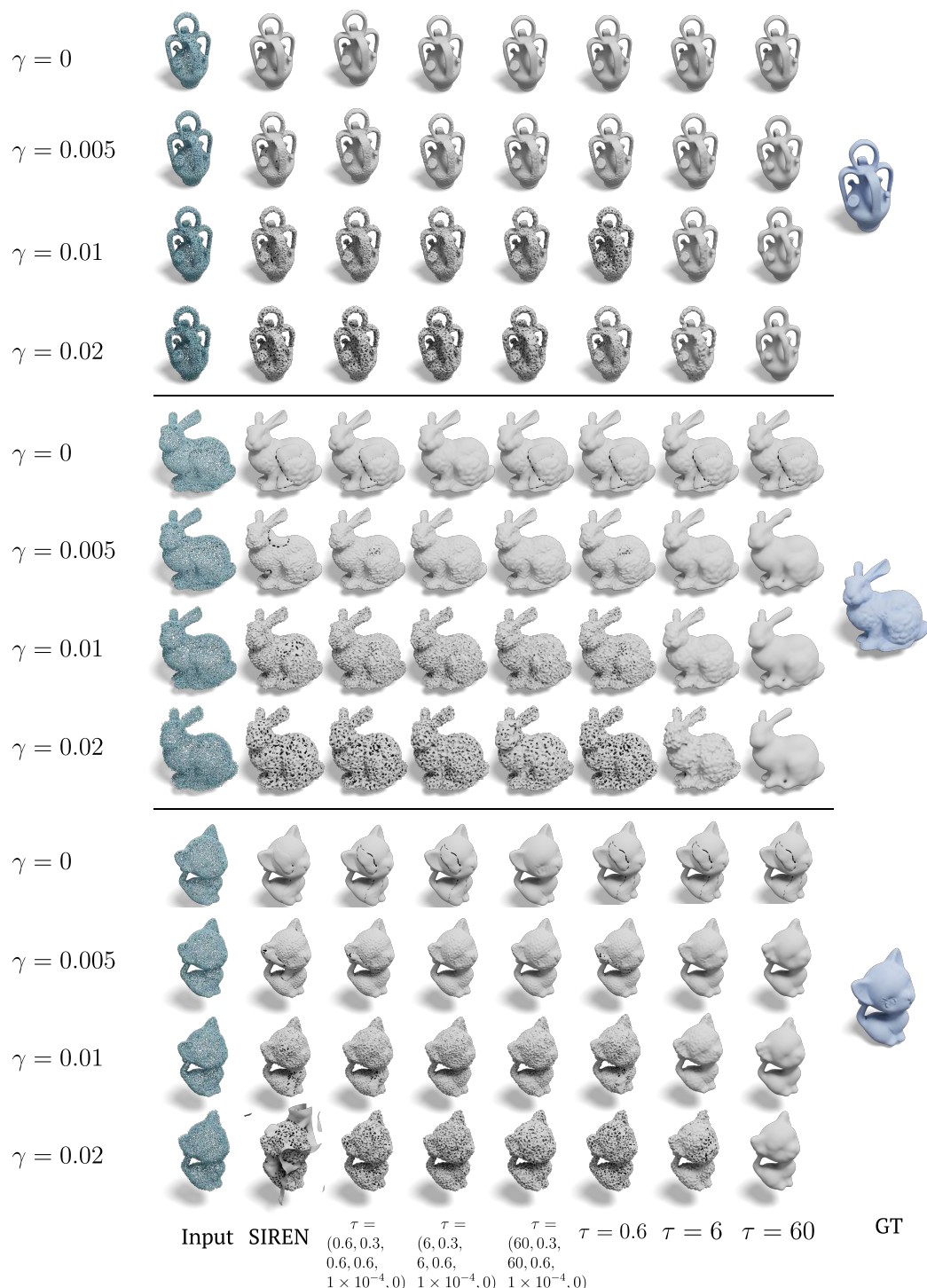

Figure 10: visual comparison on noisy input.

| | | airplane | bench | cabinet | car | chair | display | lamp | loudspeaker | rifle | sofa | table | telephone | watercraft | mean | std. |
|---|---|---|---|---|---|---|---|---|---|---|---|---|---|---|---|---|
| Normal C. ↑ | SPSR* [15] | 95.30 | 92.85 | 95.98 | 93.43 | 95.02 | 97.35 | 95.03 | 96.19 | 96.93 | 95.53 | 94.65 | 98.67 | 94.61 | 95.50 | **3.30** |
| | NSP* [30] | 86.04 | 85.37 | 91.30 | 91.08 | 87.83 | 93.70 | 90.45 | 92.95 | 94.94 | 90.18 | 87.66 | 96.87 | 91.30 | 90.74 | 5.48 |
| | SAL [1] | 77.52 | 78.04 | 90.64 | 90.32 | 79.57 | 91.74 | 86.09 | 94.60 | 86.35 | 90.44 | 77.05 | 97.59 | 87.68 | 86.69 | 9.66 |
| | IGR [11] | 74.48 | 73.98 | 88.47 | 86.00 | 75.23 | 76.79 | 83.46 | 92.61 | 78.43 | 82.07 | 73.97 | 82.29 | 83.22 | 80.85 | 11.88 |
| | SIREN [25] | 88.42 | 80.93 | 78.58 | 77.01 | 84.08 | 89.76 | 84.80 | 78.84 | 84.98 | 80.28 | 86.70 | 90.27 | 84.30 | 83.79 | 10.20 |
| | DiGS [6] | 97.19 | 93.86 | 94.61 | 93.18 | 93.98 | 97.50 | 95.59 | 96.05 | 98.12 | 96.11 | 94.18 | 99.02 | 96.28 | 95.82 | 4.44 |
| | OSP [4] | 94.97 | 91.87 | 95.23 | 92.90 | 95.44 | 97.56 | 93.00 | 95.63 | 92.62 | 95.70 | 95.90 | 97.51 | 93.23 | 94.73 | 3.94 |
| | iPSR [12] | 92.45 | 88.80 | 93.26 | 92.51 | 92.64 | 95.35 | 93.70 | 94.54 | 96.39 | 92.41 | 89.66 | 97.56 | 92.63 | 93.22 | 5.26 |
| | PGR [19] | 85.57 | 86.85 | 94.09 | 91.25 | 91.88 | 95.59 | 90.63 | 95.17 | 91.14 | 93.55 | 91.03 | 97.80 | 90.19 | 91.90 | 4.93 |
| | POCO+ [8] | 96.81 | 94.23 | **97.28** | 93.52 | 96.56 | **98.29** | 95.92 | **96.86** | 97.56 | 96.87 | 96.65 | 99.02 | 93.77 | 96.41 | 3.53 |
| | **Ours (SIREN)** | **97.82** | **94.24** | 97.06 | **94.57** | **96.63** | 96.21 | **96.52** | 92.74 | **98.47** | **97.54** | **97.38** | **99.22** | 96.36 | **96.52** | 3.31 |
| Chamfer ↓ | SPSR* [15] | 2.73 | 4.08 | 5.06 | 6.66 | 5.83 | 4.08 | 3.97 | 6.26 | 1.73 | 5.31 | 6.00 | 2.36 | 6.50 | 4.66 | 4.64 |
| | NSP* [30] | 19.44 | 7.61 | 9.57 | 6.07 | 11.75 | 7.08 | 7.33 | 13.65 | 2.45 | 8.24 | 10.14 | 3.82 | 7.96 | 8.85 | 6.96 |
| | SAL [1] | 52.97 | 45.43 | 21.19 | 14.25 | 55.97 | 23.83 | 34.35 | 13.93 | 13.33 | 17.25 | 68.27 | 6.54 | 23.29 | 29.98 | 31.86 |
| | IGR [11] | 12.44 | 69.77 | 34.50 | 24.30 | 58.89 | 91.40 | 55.51 | 19.87 | 68.71 | 46.77 | 75.63 | 83.02 | 60.22 | 62.54 | 48.44 |
| | SIREN [25] | 26.12 | 38.23 | 33.62 | 42.33 | 23.52 | 17.84 | 46.18 | 34.34 | 65.53 | 23.49 | 21.36 | 30.22 | 42.08 | 34.19 | 46.77 |
| | DiGS [6] | 2.44 | 3.87 | 8.50 | 4.82 | 7.69 | 4.45 | 3.83 | 5.95 | **1.35** | 4.50 | 6.63 | 2.36 | 3.23 | 4.59 | 4.94 |
| | OSP [4] | 5.85 | 4.40 | 6.02 | 9.28 | 5.97 | 4.20 | 11.91 | 6.63 | 9.09 | 5.09 | 5.74 | 5.15 | 9.06 | 6.80 | 6.61 |
| | iPSR [12] | 4.17 | 6.08 | 6.35 | 7.22 | 6.97 | 5.40 | 4.27 | 7.20 | 2.10 | 7.28 | 7.81 | 4.60 | 7.84 | 5.95 | 5.97 |
| | PGR [19] | 7.27 | 8.17 | 7.19 | 8.41 | 7.69 | 6.22 | 8.52 | 7.78 | 4.93 | 7.45 | 8.85 | 3.70 | 9.28 | 7.34 | 4.81 |
| | POCO+ [8] | 2.00 | 3.64 | 3.76 | 5.30 | 4.02 | **2.93** | 2.47 | **4.13** | 1.37 | 3.91 | 4.99 | 2.18 | 6.40 | 3.62 | 4.21 |
| | **Ours (SIREN)** | **1.72** | **2.92** | **3.63** | **4.61** | **3.77** | 3.64 | **2.40** | 8.46 | **1.05** | **2.97** | **3.93** | **1.84** | **3.03** | **3.38** | **3.64** |
| F-Score ↑ | SPSR* [15] | 90.17 | 76.96 | 67.50 | 69.99 | 65.21 | 81.29 | 79.09 | 56.36 | 95.66 | 70.81 | 61.00 | 94.69 | 69.64 | 75.28 | 25.76 |
| | NSP* [30] | 50.61 | 46.86 | 38.65 | 57.83 | 30.18 | 54.11 | 60.14 | 28.52 | 91.31 | 44.29 | 33.47 | 81.70 | 63.80 | 52.42 | 28.55 |
| | SAL [1] | 8.73 | 15.46 | 24.86 | 34.22 | 11.06 | 23.62 | 26.27 | 31.98 | 28.40 | 30.07 | 8.37 | 62.90 | 29.98 | 25.76 | 22.43 |
| | IGR [11] | 1.77 | 10.28 | 56.52 | 47.66 | 12.76 | 15.74 | 31.11 | 60.90 | 4.44 | 34.99 | 14.17 | 21.77 | 29.49 | 26.28 | 35.91 |
| | SIREN [25] | 45.30 | 29.34 | 17.52 | 15.28 | 39.76 | 45.30 | 42.28 | 16.07 | 44.51 | 22.96 | 26.04 | 47.63 | 27.59 | 32.34 | 30.13 |
| | DiGS [6] | 93.08 | 83.44 | 53.57 | 77.66 | 68.33 | 77.55 | 85.41 | 62.75 | 98.57 | 76.56 | 68.14 | 95.84 | 84.40 | 78.87 | 27.34 |
| | OSP [4] | 43.02 | 73.47 | 65.35 | 38.96 | 57.37 | 80.52 | 57.39 | 55.29 | 41.30 | 75.01 | 61.78 | 84.08 | 35.03 | 59.12 | 25.82 |
| | iPSR [12] | 72.59 | 62.60 | 58.69 | 66.97 | 59.01 | 73.75 | 80.01 | 53.79 | 93.43 | 62.92 | 50.46 | 91.29 | 64.98 | 68.42 | 26.36 |
| | PGR [19] | 44.70 | 42.12 | 46.98 | 57.14 | 46.95 | 57.69 | 48.46 | 50.78 | 62.57 | 43.66 | 38.29 | 84.75 | 44.64 | 51.44 | 23.12 |
| | POCO+ [8] | 96.21 | 82.62 | 81.33 | 82.85 | 83.37 | **93.08** | **93.74** | 72.98 | **99.30** | 83.13 | 71.51 | 95.31 | 74.99 | 85.42 | 33.13 |
| | **Ours (SIREN)** | **98.96** | **93.13** | **84.21** | **85.80** | **83.56** | 90.42 | 91.39 | 64.41 | 97.69 | **91.17** | **89.79** | **95.49** | **87.21** | **88.71** | 18.28 |

Table 9: Class-by-class comparison of the surface reconstruction quality on 3K-point clouds of ShapeNet [10].