# OpenReview forum: "Aligning Gradient and Hessian for Neural Signed Distance Function"
_NeurIPS.cc/2023/Conference — NeurIPS 2023 poster_

### Official Review · Reviewer_ApDg · 2023-06-10

**Soundness:** 3 good
**Presentation:** 2 fair
**Contribution:** 3 good
**Rating:** 6
**Confidence:** 4

**Summary:**

The authors propose a new smoothing loss to better regularize the estimated Signed Distance Field (SDF) from unoriented point clouds. By aligning the SDF's gradient and its Hessian, the quality of reconstructed surfaces improves significantly. Compared with commonly used Eikonal loss, the proposed loss is more effective. Experiments validate the effectiveness of the proposed method as well as not so sensitive to input noise and hyper-parameters.

**Strengths:**

1. The proposed loss is novel and theoretically reasonable.
2. The loss could be easily worked with different methods as a plugin.
3. The experiments are almost complete and big improvements validate the effectiveness.

**Weaknesses:**

The paper is poorly written. The authors write the paper more like from a mathematic point of view rather than computer vision point of view. Many details are missing, and some said "obviously" and "evidence" are not clear at all. For a computer vision paper, Figures are much more important for illustration and detailed explanation of Figures and Equations, especially where to look at in the figure and why an equation intuitively works.

However, if the authors could provide details in the rebuttal process and the final version, I believe that the paper could be accepted. If not, it should be rejected, as this is not a math paper.

**Questions:**

1.	The first paragraph in Sec 3.3 requires figure illustration. It’s difficult to understand why Equ. 4 intuitively works. What is Equ. 4 enforcing? What do you mean by "alignment"? A few figures are necessary. It is not clear at all where Equ. 4 comes from.
2.	In Figure 2, The 100 points inputs are different for three methods? Please show the same inputs’ results as comparisons; otherwise, it is  not convincing.
3.	In Figure 3, it is NOT evidence that the alignment loss can suppress the ghost geometry as mentioned in Line 173. The authors need clearly show which region in Figure 3 is problematic, and why the last column is good. It seems the alignment loss works similar to Dirichlet loss, and I do not see any benefit. Are the black lines zero-isosurface? Then the last column’s result seems quite bad.
4.	The big Q in Equ. 7 is not defined.
5.	Table 5’s caption needs explanations of each row, especially the difference between same name’s rows.


**Limitations:**

The current experiments only show surface reconstruction from point clouds. But I think the loss could benefit other scenarios, like incorporated into Neus, but the authors do not show its effectiveness. Also, whether the loss works for unsigned distance field (UDF) also? This could be an interesting future topic. The current point cloud scenario is somewhat limited.

---

> ### Author Rebuttal · Authors · 2023-08-09
>
> We thank the reviewer for considering our method novel and theoretically reasonable.
>
> **Q1: Explanation of Sec 3.3 and Eq. 4.**
>
> We are sorry for the unclear writing. We will further improve the readability in the revision. Recall that one of the basic properties of SDF is the Eikonal condition (line. 116) $\lVert\nabla f_\theta\rVert_2^2=1$, we can differentiate both sides of the identity and we can get the eigenfunction of the Hessian matrix (Eq. 2): $\mathbf{H}\_{f\_\theta} \nabla f\_\theta \(\boldsymbol{q}\)=\mathbf{0}$, which implies the gradient of SDF is one of the eigenvectors of the Hessian matrix of SDF $f$ and its corresponding eigenvalue is 0. The more detailed proof is available in ref. [30]. The property implies that the gradient must align with one eigenvector of the Hessian matrix with 0 eigenvalue. Our regularization term (Eq. 4) is exactly based on this fundamental property. Therefore, we use "alignment" to enforce the gradient to become one of the eigenvectors of the Hessian matrix, with the corresponding eigenvalue being 0 that is the second derivative in the gradient direction is zero. Experimental results show that the alignment requirement is effective, i.e., not only effectively suppresses the ghost geometry but also preserves geometric details.
>
>
>
> **Q2: The issue of Fig. 2**
>
> Thanks for pointing this out!
>
> We follow the 2D toy problem settings as DiGS which samples 100 points from analytical equations at each epoch. So it seems the points leveraged by different methods are different. We agree that it is misleading and we conducted experiments with fixed points in **Fig. 1 in the PDF file**, which is used to replace Fig. 2.
>
> **Q3: The issue of Fig. 3**
>
> We are sorry for this issue. We will replace it with our new figure (**Fig. 2 in the PDF file**) to demonstrate our benefits.
>
> **Q4: The definition of Q**
>
> Q is the set of sampled points and we defined it in **Sec. A in the supplementary material**.
>
> We follow IGR [ICLR 2020] to sample point clouds and directly leverage the sampled points produced by NeuS on Ray for multi-image input.
>
> Specifically, suppose that $p_i$ is a point in the input $P$, we define the Gaussian function rooted at $p_i$ and take the distance to its $k$-th nearest neighbor ($k = 50$ by default follow IGR) as the standard deviation. Then we sample points from each distribution.
>
> **Q5: Explanation of Table 5**
>
> Thanks for the question. In the paper, all methods marked with '*' means it leverages with normals in the training stage, and those marked with '+' means this is a supervised one. We have compared different optimization-based methods under DFAUST and the different symbols mean its training settings.
>
> In all, our method can learn shape space without requiring input normals or additional supervision but still produces more faithful shapes than others.
>
> We will revise it following your advice.
>
> **Q6: Extension of our method**
>
> Thanks for your constructive comments! It is interesting to explore our method with multi-image input and fit UDF.
>
> We have incorporated our loss into NeuS and conducted the experiments under the DTU dataset in **Sec.D.5 in the supplementary material** (we enclosed the experimental results in the submission phase). It can be seen that our method supports multi-image inputs and effectively improves the quality of results compared to the original NeuS.
>
> For UDF, unfortunately, it does not satisfy the Eikonal term since the norm of gradient is 0 on the 0-isosurface. Therefore, the key alignment property does not hold anymore and thus our algorithm may not work well for fitting UDF.

---

### Official Review · Reviewer_csEg · 2023-07-02

**Soundness:** 2 fair
**Presentation:** 2 fair
**Contribution:** 3 good
**Rating:** 5
**Confidence:** 5

**Summary:**

The paper under review proposes a method for signed distance function (SDF) reconstruction from point cloud data without the use of normal data at the points.  The paper proposes a new term as a regularization to complement the EIkonal loss, which is a necessary condition for an SDF.  The purpose of this regularization is to reduce "ghosting" effects.  The term is obtained by differentiating the Eikonal constraint, resulting in a second-order constraint, namely that the Hessian times the gradient is zero.  The paper shows experimental comparison to SOA on multiple benchmarks with favorable results.

**Strengths:**

- New regularization term for neural SDFs; original to best of my knowledge; neural SDFs are a very relevant research topic of interest in vision/graphics
- Paper is mostly clear
- Experiments on many benchmark datasets, with favorable results to the authors' approach

**Weaknesses:**

- Experimental results on SRB and ShapeNet use different metrics than recent papers in area, e.g., DiGS[6] - not clear why they have been changed.  Please report results with previous protocols.
- A new sampling strategy is reported in supplementary for the alignment loss.  Not clear what the effect of this is, and why it is needed.
- Theoretical justification is weak, e.g., line 156-174; there is no clear understanding of this new term - no clear explanation of why the ghosting is reduced with this term.
--The authors attribute ghosting to in practice only applying the Eikonal constraint to a sampling of points; unclear if this is true due to random sampling or this is only part of the problem.  The Eikonal equation does not have a unique solution which is in part why there is ghosting.
--Unclear why the smoothness term not being min zero means it's difficult to regulate.
--It is stated that the norm and direction of the gradient is de-coupled.  However, a minimization of the loss is being considered - the optimizing flow for both the Eikonal and alignment will both change the direction & norm of the gradient during minimization.
- Line186-187: the constraint is true a.e. so why is the adaptive weighting needed?

**Questions:**

Address the weaknesses.

**Limitations:**

Not discussed.

---

> ### Author Rebuttal · Authors · 2023-08-09
>
> Thanks for the valuable comments. We will incorporate all feedback into the revised version.
>
> **Q1: Different metric compared to DiGS**
>
> To our knowledge, there are two commonly used evaluation settings under SRB, one used by DiGS [CVPR 2022] and the other proposed by Shape as Points [NIPS 2021].
>
> The version of DiGS operates in the resolution of $512^3$ and uses Chamfer distance and Hausdorff distance at the original scale, while the version of Shape as Points operates in the resolution of $256^3$ and uses Chamfer distance, F-Score, and Normal Consistency, but it does not give the number of evaluation points. Therefore, we sample 100K evaluation points and follow the settings of Shape as Points in this paper. Following your advice, we also show the results under the settings of DiGS, see **Tab. 3 in the global response box**.
>
> It can be seen that our method outperforms DiGS in terms of Hausdorff distance. To summarize, our method consistently outperforms DiGS, in terms of whether Hausdorff distance or for F-Score (indicated in the main paper).
>
>
>
> **Q2: Effect of sampling strategy for alignment**
>
> The sampling strategy is not new. Actually, we use the sampling strategy following IGR [ICLR 2020] and NeuralPull [ICML 2021], to sample more points near the input points rather than uniformly the bounding box.
>
> The rationale behind this lies in that the SDF only guarantees to be differentiable in a narrow region nearby the surface (considering faraway surface points which have two or more same minimal distance that cannot be differentiable). Besides, for the reconstruction methods, the 0-isosurface is of more interest than other level sets. We compared different sampling strategies as **Tab. 4 in the global response box** (We disable adaptive weight here to validate sampling strategies only). It can be seen that sampled points near the surface are better than samples randomly on the bounding box.
>
> **Q3: Understanding of ghost geometry and our term**
>
> Thanks for pointing out this issue. As you observed, the Eikonal term has many invalid solutions (with ghost geometry) that cannot be filtered out on finite and discrete points, as discussed in DiGS. By contrast, second-order information can effectively limit the solution space.
>
> Secondly, the parameters of the network are typically larger than input points, leading to over-parameterized neural networks with an extensive array of parameters, which can further convolute the optimization process, as described by Empirical Analysis of the Hessian of Over-Parametrized Neural Networks [Sagun et al., 2017].
>
> Our regularization term is based on the fundamental geometry property of SDF to limit the solution space and reduces the number of possible solutions. The occurrence of ghost geometry contradicts the fundamental property of SDF. We construct a similar toy problem as DiGS with the shape 'L' with 100 points to demonstrate the effect of our term. We set the weighting coefficient of all regularization terms to 100 for fairness (the default coefficient used by DiGS) shown in **Fig. 1 in the PDF file**. The experimental results validate our observation. (Note that DiGS leverages 15K points in its main paper for the toy example.)
>
> To further validate the effect of our term, we increase the weighting coefficient of the regularization term to 1e3, 1e5, and 1e7, respectively, as shown in **Fig. 2 in the PDF file**. It can be seen that even with large weight, our regularization term consistently yields high-fidelity surfaces without introducing over-smooth and ghost geometry. However, smoothness energy terms easily degenerate.
>
> In essence, other general smoothness energies, which are not always to zero, tend to force the derivative to 0 and will be smoothing in unwanted ways. Instead, our regularization term is more relaxed to avoid over-smooth results, which enforces the gradient to become one of the eigenvectors of the Hessian matrix, with the corresponding eigenvalue being 0 that only the second derivative in the gradient direction being zero satisfies the fundamental property of SDF. To illustrate the feature-preserving ability of our method, we conducted experiments under a lion shape with 100K points as input. As shown in **Fig. 3 in the PDF file**, our method not only suppresses ghost geometry but also recovers high-fidelity geometric details. However, smoothness energy terms tend to produce over-smoothed results. Finally, we agree that the norm and direction of the gradient cannot be totally de-coupled. We shall make it more rigorous in this revision.
>
>
>
> **Q4: Effect of adaptive weighting**
>
> As discussed in #Q2, it is better to emphasize the points nearby the underlying surface. We conducted an ablation study about how adaptive weighting influences the reconstruction result; **Tab. 7 in the main paper** demonstrates that an adaptive weighting scheme ($\delta = 0$ means disabling it with the adaptive weighting constantly is 1) a better strategy.

---

> > ### Comment · Reviewer_csEg · 2023-08-18
> > **Response to rebuttal**
> >
> > I thank the reviewers for the rebuttal.  It addressed my main concern with different metrics compared to DiGs.  I think this paper is a good contribution with good empirical evidence, but still a bit lacking in the theory.  I have increased my rating.

---

> > > ### Author Response · Authors · 2023-08-18
> > > **Thanks for your response**
> > >
> > > Thanks for your quick response.
> > > We really appreciate your decision to increase the rating. At the same time, we shall follow your advice to further consolidate the theory. Thanks!

---

### Official Review · Reviewer_JRpq · 2023-07-05

**Soundness:** 3 good
**Presentation:** 3 good
**Contribution:** 3 good
**Rating:** 6
**Confidence:** 3

**Summary:**

The authors provide a novel regularizing term for training of implicit neural SDF representations, motivating the function gradient to be a zero eigenvector of the Hessian near the zero level set (the surface). They demonstrate that it can be used to reduce the appearance of ghost geometry and increase quality of the representations.

**Strengths:**

- A principled novel regularization term that seems to significantly improve the learned representations.
- Is tackling an important application domain, as neural geometry representations are broadening the application domain of deep learning techniques.

**Weaknesses:**

- It seems unclear to me how the method is specifically designed to handle unoriented point clouds. If there is a claim that the regularizer somehow motivates consistent alignment of resulting normals, it'd be nice to have some commentary on this.
- The evaluation metrics are a little strange to me:
   - It's unclear how they are comparing normals, as there would need to be a correspondence between the reconstructed surface and the ground truth surface.
   - F-score seems a bit of an odd choice. I initially assumed that they are using the interior of the isosurface to specify points classified as inside the volume, but the default threshold 0.005 seems quite low for this. So perhaps they are asking for the network output to classify points on the shape, but I am not sure of how they do this either, as nearly all points will evaluate to nonzero values.
  - In my opinion, Hausdorff would be good to include, as it would give a sense of the maximum error.

**Questions:**

See above weaknesses.

**Limitations:**

The authors adequately address the limitations of the methods, which are mostly that there is still room for further improvement in their output.

---

> ### Author Rebuttal · Authors · 2023-08-09
>
> We appreciate that the reviewer finds our paper novel. We address additional comments below.
>
> **Q1: How to handle unoriented point clouds?**
>
> Yes, our alignment term encourages consistent orientations of normal vectors. Based on our $C^{\infty}$ activation function (sine and softplus), when we request the Hessian matrix to degenerate along the eigenvector corresponding to the zero eigenvalue, that is the second derivative in the gradient (normal) direction is zero, the degeneration direction of the Hessian matrix in the neighborhood also tends to be consistent, leading to consistent normals.
>
>
> **Q2: Confusion about the evaluation metric**
>
> In the paper, we used the commonly used metrics for evaluating reconstruction methods including Chamfer distance, F-Score, and Normal consistency proposed by OccupancyNet [CVPR 2019] and Convolutional Occupancy Networks [ECCV 2020]. We give their definitions in **Sec. C in the supplementary material**.
>
> For the purpose of comparing normals, we sample the same number of points from the reconstructed surface and the ground-truth surface, respectively. For each sample point on the reconstructed surface, we find the nearest point in the other point set and keep the dot-product of the two normals. Note that the computation is bi-directional.
>
> The original F-Score is defined as the harmonic mean between precision and recall. To define the F-Score in our scenario, we sample the same number of points from the reconstructed surface and the ground-truth surface, as mentioned above. For recall, it counts how many points on the GT mesh lie within a certain distance (threshold t) to the reconstruction surface. For precision, it counts the percentage of points on the reconstructed mesh that lie within a certain distance (threshold t) to the GT. The F-Score is then defined as the harmonic mean between precision and recall, see the definitions of precision and recall in **Sec. C in the supplementary material**
>
> Furthermore, we agree that Hausdorff distance is a good choice. We compared ours and DiGS using Chamfer distance and Hausdorff distance under SRB, see **Tab. 2 in the global response box**.
>
> We will add it in the revision.

---

> > ### Comment · Reviewer_JRpq · 2023-08-15
> > **Thank you for the clarifications**
> >
> > After reading the other reviews and rebuttals, I'd like to keep my score where it is.

---

> > > ### Author Response · Authors · 2023-08-16
> > > **Thanks for your reply**
> > >
> > > We appreciated that your decision to keep your score. It’s glad we clarified your doubts and give you a better view about our work.
> > > We will follow your advice and revise our paper.

---

### Official Review · Reviewer_krE5 · 2023-07-06

**Soundness:** 2 fair
**Presentation:** 1 poor
**Contribution:** 3 good
**Rating:** 5
**Confidence:** 5

**Summary:**

The paper considers the task of surface reconstruction from point clouds without normals and uses neural signed distance functions. They consider previous smoothness losses, and come up with their own by taking the derivative of the Eikonal equation, thus guaranteeing that their loss constrains a fundamental property of SDFs. They motivate this loss in various other ways (which to me are fairly unclear). They also show good results on many datasets, and compare to many different types of approaches.

**Strengths:**

- Using second order information derived from the Eikonal equation makes sense
- Lots of experiments and comparisons to many types of methods

**Weaknesses:**

- One of the benefits of the split of the ShapeNet dataset is that you use is that you can compute IoU with it, which is a very different but important measure to IoU.
- Your explanation for why L_{align} is different from the Eikonal term is essentially that it works better for finitely many points, which is not an explanation. You should explain that with finitely many points constraining second order information reduces the number of possible solutions, and maybe allude to the toy problem in DiGS (or even construct the same toy problem with your loss term).
- Your explanation for why not smoothness energy is not very compelling either. Your main argument is that other energies are not 0 at the optimum, hence it is hard to choose how regularisation to apply. A much better argument is your earlier point that the loss is derived from a fundamental property of SDFs, while the others are general smoothing losses that are not specifically guiding towards a proper SDF and will be smoothing in unwanted ways.

Overall great idea and results, not great presentation/soundness. Your explanation for your loss needs to be more clear, keeping it to it is a fundamental property of an SDF works just fine. A toy problem to back this up could be to see what happens when you regularise the 2D shapes with high weight with the different smoothness energies, mostly likely other methods would stop obeying the points and go for a smoother surface while yours would still obey the input points.

**Questions:**

- Is there a reason you are using most of DiGS' datasets but not using their version of the metrics?
- Your explanation of the implication is Rodigues' formula is not clear. I couldn't find the formula name in your citation [34], and looking else online it seems that the formula states that one of the eigenvalue-eigenvector pairs of the Hessian should be a principle curvature, principle direction pair, not all eigenvalue-eigenvector pairs? Why does it imply equation (3)?
- What is Figure 3 trying to show us?

**Limitations:**

Not discussed.

---

> ### Author Rebuttal · Authors · 2023-08-09
>
> We thank the reviewer for constructive feedback. In the following, we will address the main concerns carefully and seriously.
>
> **Q1: Compute IoU under ShapeNet**
>
> Following your advice, we report the IoU statistics under ShapeNet, see **Tab. 1 in the global response box**. At the same time, we will add the statistics to the paper.
>
> It can be seen that our method is a bit inferior to the supervised method POCO (note that ours outperforms POCO in terms of Normal Consistence, Chamfer distance, and F-Score, as shown in the main paper). However, our method is consistently better than the other optimization-based methods.
>
> **Q2: Compared to the Eikonal term and other smoothness energy terms**
>
> Thanks for pointing out this issue. As you observed, when one enforces the Eikonal term on finite and discrete points, still a large number of invalid solutions (with ghost geometry) cannot be filtered out. By contrast, second-order information can effectively limit the solution space.
>
> Secondly, the parameters of the network are typically larger than input points, leading to over-parameterized neural networks with an extensive array of parameters, which can further convolute the optimization process, as described by Empirical Analysis of the Hessian of Over-Parametrized Neural Networks [Sagun et al., 2017].
>
> As for various smoothness energy terms (i.e., Laplacian energy used by DiGS), they have a negative effect (missing geometric details) although they can reduce the number of possible solutions. In other words, the ability to accurately represent geometric details is greatly diminished. By contrast, our regularization term is based on the fundamental geometry property of SDF, which doesn’t enforce unwanted smoothness. We construct a similar toy problem as DiGS with the shape 'L' with 100 points to demonstrate the effect of our term. We set the weighting coefficient of all regularization terms to 100 for fairness (default coefficient used by DiGS), shown in **Fig.1 in the PDF file**. The experimental results validate our observation. (Note that DiGS leverages 15K points in its main paper for the toy example.)
>
> To further validate the effect of our term, we increase the weighting coefficient of the regularization term to 1e3, 1e5, and 1e7, respectively, as shown in **Fig. 2 in the PDF file**. It can be seen that even with large weight, our regularization term consistently yields high-fidelity surfaces without introducing over-smooth and ghost geometry. However, smoothness energy terms easily degenerate.
>
> In essence, other general smoothness energies, which are not always to zero, tend to force the derivative to 0 and will be smoothing in unwanted ways. Instead, our regularization term is more relaxed to avoid over-smooth results, which enforces the gradient to become one of the eigenvectors of the Hessian matrix, with the corresponding eigenvalue being 0 that only the second derivative in the gradient direction being zero satisfies the fundamental property of SDF. To illustrate the feature-preserving ability of our method, we conducted experiments under a lion shape with 100K points as input. As shown in **Fig. 3 in the PDF file**, our method not only suppresses ghost geometry but also recovers high-fidelity geometric details. However, smoothness energy terms tend to produce over-smoothed results.
>
> **Q3: Different metrics compared to DiGS**
>
> To our knowledge, there are two commonly used evaluation settings under SRB, one used by DiGS [CVPR 2022] and the other proposed by Shape as Points [NIPS 2021].
>
> The version of DiGS operates in the resolution of $512^3$ and uses Chamfer distance and Hausdorff distance at the original scale, while the version of Shape as Points operates in the resolution of $256^3$ and uses Chamfer distance, F-Score, and Normal Consistency, but it does not give the number of evaluation points. Therefore, we sample 100K evaluation points and follow the settings of Shape as Points in this paper. Following your advice, we also show the results under the settings of DiGS, see **Tab. 2 in the global response box**.
>
> It can be seen that our method outperforms DiGS in terms of Hausdorff distance. To summarize, our method consistently outperforms DiGS, in terms of whether Hausdorff distance or for F-Score (indicated in the main paper).
>
> **Q4: Misleading of Rodrigues’ formula**
>
> We are sorry for this misleading, it is actually not related to Rodrigues’ formula. A more suitable explanation and reference is as follows ref. [30]: See Chapter 2. When the situation reduces to 2D, it coincides with your observation - one of the eigenvalue-eigenvector pairs of the Hessian should be a principle curvature, principle direction pair. We will fix this in the revision.
>
> **Q5: The objective of Figure 3**
>
> Thanks for pointing out the misleading figure. We shall replace it with a more informative figure (**see Fig. 1 in the PDF file**).

---

> > ### Comment · Reviewer_krE5 · 2023-08-16
> > **Thank you for your clarifications. I still have some concerns on the IoU results.**
> >
> > Thanks for giving the results on ShapeNet for IoU. POCO performing better is not a concern, it is a supervised method and thus is an unfair comparison. However, the actual IoU values are very suspicious to me. The values for IoU vary greatly from the DiGS paper: in that paper the mean IoU is 0.939 whereas you report 78.44? How are you calculating the IoU?
> >
> > The Fig 1 and Fig 2 in the PDF file is a much better representation of what is going on. It would be good if you have at least one of them in the main paper.
> >
> > Thanks for providing the metrics used in DiGS (it was actually started by Neural Splines in CVPR2021).
> >
> > In regards to Section 3.2, do you really need that the eigenvectors define the principal directions? By taking the derivative of the eikonal equation you get (2), which is what you use in practice. I don't understand why the principal directions are important and/or how they are used in the paper?

---

> > > ### Author Response · Authors · 2023-08-16
> > > **Thanks for your questions. We hope that our answers can address your concerns.**
> > >
> > > Thanks for your questions. We hope that our answers can address your concerns.
> > >
> > > **Question about IoU**
> > >
> > > For the computation of IoU, we adopt the evaluation code from POCO, which is derived from a widely used IoU evaluation code by ConvOccupancyNet[ECCV 2020]. This code retrieves the volume data (occupancies) from 'points.npz' and computes the IoU using 100K evaluation points. In contrast, DiGS employs its own evaluation code (not popular actually).
> > >
> > > In their main paper, DiGS does report superior IoU values. However, its performance degrades for some categories with concave features (See Tab.1 and Fig. 1 in the supplementary material), such as lamps. In fact, the inconsistency is due to different experimental settings: DiGS directly utilizes 100K points as input to conduct the experiment, while our experiments use only 3K points. Hope that our answer can address your concern.
> > >
> > > **Question about eigenvectors**
> > >
> > > We do not employ the eigenvectors corresponding to principal directions in our paper, but it is insightful to understand the relationship between the Hessian matrix of SDF and the surface. When we enforce that the gradient aligns with the kernel space (the space spanned by the eigenvector corresponding to 0), the other two eigenvectors naturally align with the principal directions. While directly constraining the direction of the principal curvature of the surface might produce interesting results, our method doesn't impose any constraints on the curvature and related directions, that’s why our approach has a better ability to recover geometric details. Thanks.

---

> > > > ### Comment · Reviewer_krE5 · 2023-08-18
> > > > **Still not happy with IoU, but my other concerns have been address. Increased my score by 1, but hope to hear back about my concerns.**
> > > >
> > > > The ShapeNet data is actually from the Occupancy Networks paper (not the followup, ConvOccNet). That data gives inside-outside annotations for 100k points, so it is very weird to only use 3k when you have access to all 100k points. Regardless I have checked and both Occupancy Networks and ConvOccNet say they use all 100k points for IoU. Also note that DiGS specifically mentions they are using the evalusation procedure from Neural Splines, who use the IoU methodology of Occupancy Networks.
> > > >
> > > > POCO does say that they only use 3k points, however they report a IoU of 0.926 on ShapeNet, which sounds much more realistic. On the other hand, your have reported 84.25 for POCO, which is contradictory!!! Even if it is tested on 3k points, I fail to see how DiGS would get a mean IoU of 78.44 and especially how NSP (which uses normals) would get a mean IoU of 67.34 (unless the 3k subset is not very random)!!
> > > >
> > > > Thanks for the clarification about the eigenvectors. Given that all my concerns other than the IoU results have been addressed, I am happy to increase my score from 4 to a 5: I liked the idea in the paper, and the results (though I am a bit skeptical on them now after these IoU results) seem very good, my issues were with the writing rigour which seem like they will be cleaned up.
> > > >
> > > > I hope the authors can still clarify my doubts with the IoU results. I also hope the authors will release their code and evaluation code to allow other to double check their results.

---

> > > > > ### Author Response · Authors · 2023-08-20
> > > > > **Thanks for your patience and focus**
> > > > >
> > > > > Thanks for your patience and focus.
> > > > >
> > > > > **About IoU**
> > > > >
> > > > > In response to your concern, we have carefully double-checked our evaluation code and occupancy data from 'points.npz'. We found the ground truth occupancy data is not queried from the standard [-0.5, 0.5] space, but instead had a slight offset. However, a normalization operation was applied to the reconstruction meshes for IoU computation, which likely caused the confusion you pointed out.
> > > > >
> > > > > We apologize for our mistake. We have resolved this issue by aligning the results with the ground truth occupancy data and reevaluating the IoU:
> > > > >
> > > > > |                     |                         | $\text{SPSR}^*$ | $\text{NSP}^*$ | $\text{SAL}$ | $\text{IGR}$ | $\text{SIREN}$ | $\text{DiGS}$ | $\text{OSP}$ | $\text{iPSR}$ | $\text{PGR}$ | $\text{POCO}^{+}$ | $\textbf{Ours (SIREN)}$ |
> > > > > |---------------------|-------------------------|-----------------|----------------|--------------|--------------|----------------|---------------|--------------|---------------|--------------|-------------------|-------------------------|
> > > > > | $ \text{IOU}$ | $\text{mean}\uparrow$   | $0.9058$        | $0.7908$       | $0.6062$     | $0.3333$     | $0.4114$       | $0.8954$      | $0.5084$     | $0.8501$      | $0.7834$     | $\textbf{0.9440}$ | $0.8980$                |
> > > > > |                     | $\text{std.}\downarrow$ | $0.0677$        | $0.1241$       | $0.1861$     | $0.2451$     | $0.1582$       | $0.0883$      | $0.1256$     | $0.0934$      | $0.1117$     | $\textbf{0.0555}$ | $0.0647$                |
> > > > >
> > > > > Our code, evaluation code, and detailed IoU statistics table are available after this revision(seems that we are not allowed to share external links).
> > > > >
> > > > > It is worth noting that for other metrics, we apply normalization to both the reconstruction and the ground-truth surfaces to ensure alignment.
> > > > >
> > > > > In terms of the number 3K, it’s the number of single input point cloud.
> > > > >
> > > > > During the IOU evaluation, we consistently use 100K evaluation points for evaluating the performance on all methods’ result meshes.
> > > > >
> > > > > **Neural Splines are sensitive to hyper-parameters?**
> > > > >
> > > > > A: In their paper, both the number of input points and the number of Nyström samples are 1K. In our paper, we use 3K input points with 3K Nyström samples. Yes, we found that their results are not as good as that in their paper. There are some artifacts emerged such as the surface sheets far away from the surface. This shows that Neural Splines are sensitive to hyper-parameters. (We double-checked our code, and have confidence about our code.)
> > > > >
> > > > > Thank you once again, we really appreciate your efforts. Your careful review greatly benefits the community.

---

> > > > > > ### Comment · Area_Chair_Bags · 2023-08-20
> > > > > > **Please remove the external link**
> > > > > >
> > > > > > Dear authors,
> > > > > >
> > > > > > Per the discussion policy, I am sorry to tell you that you can not post external links in the discussion. Please remove the links as soon as possible from all your responses to avoid any consequences.
> > > > > >
> > > > > > Thanks,
> > > > > >
> > > > > > AC

---

> > > > > > > ### Author Response · Authors · 2023-08-21
> > > > > > > **Sorry for the delay**
> > > > > > >
> > > > > > > We have removed the external link.

---

### Official Review · Reviewer_XXzk · 2023-07-07

**Soundness:** 3 good
**Presentation:** 3 good
**Contribution:** 3 good
**Rating:** 6
**Confidence:** 4

**Summary:**

This paper introduces a novel approach to learning the SDF directly from point clouds without the use of surface normals. The key insight behind this is that aligning the gradient of the SDF with its Hessian allows for better training of the distance field. Through extensive experiments, the paper demonstrates the effectiveness of the proposed approach in accurately recovering the underlying shape and effectively reducing geometry artifacts.



**Strengths:**

- The paper exhibits a good writing style, effectively conveying the ideas and concepts to the readers. The content is clear, concise, and well-structured, allowing for easy understanding of the proposed method.

- The paper introduces a novel method for learning the signed distance function directly from unoriented point clouds.

- The paper presents extensive experimental results that showcase the effectiveness of the proposed approach.



**Weaknesses:**

- The paper does not explicitly discuss potential limitations or challenges of the proposed method. Addressing and acknowledging these limitations would provide a more comprehensive understanding of the method's applicability and potential areas for improvement.

- The paper lacks a time analysis of the proposed method. While the experimental results demonstrate the effectiveness of the approach, the paper does not provide insights into the computational efficiency or time complexity of the method. Understanding the time requirements would be valuable for assessing its practicality and potential for real-time applications.

**Questions:**

Can the proposed method guarantee the reconstruction of watertight surfaces? For large-scale scenes, are the reconstructed surfaces watertight?

**Limitations:**

The paper lacks a discussion of potential limitations and the negative social impact of the proposed method.

---

> ### Author Rebuttal · Authors · 2023-08-09
>
> We sincerely thank the reviewers for their appreciation and invaluable comments. We will incorporate all feedback into the revised version.
>
> **Q1: Limitations and negative social impact**
>
> We shall point out the following limitations in this revision: (1) it is weak in handling sparse input such as sketch point clouds or LiDAR. In particular, most LiDAR data in the KITTI dataset are partial scans. This presents significant challenges in closing the gaps between the stripes and completing large missing parts. We enclosed the relevant experimental results in the **PDF file** (**Fig. 4**) and will include them in the supplementary. (2) Its potential negative social impact includes unauthorized replication of mechanical designs and intentional content creation as other reconstruction methods.
>
> **Q2: Time analysis of the method**
>
> We made a detailed comparison with IGR, SIREN, and DiGS on the timing cost and the number of parameters for optimizing for a single point cloud in **Sec.B.2** and **Tab. 3** in the supplementary material. See the table below for more information.
>
> |            |$\text{IGR}$|$\text {SIREN}$|$\text{DiGS}$|$\text {Ours}$|
> |:------------|---------|:---------:|:---------:|:---------:|
> |#$\text{parameters}$|$\text{1.86M}$|$\text{264.4K}$|$\text{264.4K}$|  $\text{264.4K}$|
> |$\text {time[ms]}$|$\text{50.73}$|$\text{11.52}$|$\text{36.28}$|$\text {40.10}$|
>
> The optimization time statistics are made on a single GTX 3090 GPU. It can be seen that the timing costs of DiGS and ours are higher than the conventional SIREN since DiGS and ours need a second-order optimization. However, ours is more computationally efficient than IGR.
>
> **Q3: Guarantee about the watertight reconstruction surface?**
>
> Since our approach fits the underlying SDF, which naturally guarantees that the reconstruction surface (the 0-isosurface) is watertight. But it must be pointed out that when the points are too sparse or with a lot of missing parts, the produced surface may consist of many separate connected components. See **Fig.4 in the PDF file**. Furthermore, for large-scale scenes (such as from Matterport3D), it is extremely hard to represent large-scale scenes within a single network due to the catastrophic forgetting issue of the neural networks. The sliding window strategy proposed by recent works (e.g. DeepLS [ECCV 2020] and BlockNeRF [CVPR 2022]) seems helpful. We shall list it as an interesting future work. Thanks.

---

> > ### Comment · Reviewer_XXzk · 2023-08-20
> > **Comment by Reviewer XXzk**
> >
> > Having reviewed the rebuttal and taken into account other assessments, the provided response has addressed my concerns. I appreciate the author's feedback.

---

> > > ### Author Response · Authors · 2023-08-20
> > > **Thanks for your reply**
> > >
> > > Thank you for taking the time to review the rebuttal and other assessments. We are pleased to see that the provided response has effectively addressed your concerns. Your feedback is greatly appreciated as it helps refine the quality of our work.

---

### Author Rebuttal · Authors · 2023-08-09

We sincerely appreciate all valuable comments and suggestions, which helped us to improve the quality of the article. We have carefully considered all your advice and addressed the questions raised, see details in the separate response and the attached PDF files. Specifically:

- We clarify the motivation and insight of the method compared to the Eikonal term and other general smoothness energy terms. Moreover, we conducted 2D experiments compared to them (the black bold line represents 0-isosurface), see the figures in the attached PDF files.

- We report the IoU under ShapeNet:


|                      |                         | $\text{SPSR}^*$ | $\text{NSP}^*$ | $\text{SAL}$ | $\text{IGR}$ | $\text{SIREN}$ | $\text{DiGS}$ | $\text{OSP}$ | $\text{iPSR}$ | $\text{PGR}$ | $\text{POCO}^{+}$ | $\textbf{Ours (SIREN)}$ |
|:--------------------:|:-----------------------:|:---------------:|:--------------:|:------------:|:------------:|:--------------:|:-------------:|:------------:|:-------------:|:------------:|:-----------------:|:-----------------------:|
| $\text{IOU}$ |  $\text{mean}\uparrow$  |     $81.01$     |     $67.34$    |    $48.89$   |    $30.28$   |     $35.65$    |    $78.44$    |    $57.86$   |    $75.53$    |    $69.26$   |  $\textbf{84.25}$ |         $81.19$         |
|                      | $\text{std.}\downarrow$ |     $15.10$     |     $21.85$    |    $30.93$   |    $33.46$   |     $25.96$    |    $18.72$    |    $27.28$   |    $19.78$    |    $20.22$   |  $\textbf{13.62}$ |         $16.37$         |

$\quad\quad\quad\quad\quad\quad\quad\quad\quad\quad\quad\quad\quad\quad\quad$ Table 1: IoU under ShapeNet

- We compare with DiGS under SRB using the evaluation settings of DiGS:

||$\text {Mean}$||$\text {Anchor}$||||$\text{Daratech}$||||$\text{DC}$||||
|:---:|:---:|:---:|:---:|:---:|:---:|:---:|:---:|:---:|:---:|:---:|:---:|:---:|:---:|:---:|
||$\text{GT}$||$\text{GT}$||$\text{Scans}$||$\text{GT}$||$\text { Scans }$||$\text{GT}$||$\text{Scans}$||
|$\text{Method}$|$d_C$|$d_C$|$d_C$|$d_H$|$d_{\vec{C}}$|$d_{\vec{H}}$|$d_C$|$d_H$|$d_{\vec{C}}$|$d_{\vec{H}}$|$d_C$|$d_H$|$d_{\vec{C}}$|$d_{\vec{H}}$|
|$\text{DiGS}$|$\textbf{0.19}$|$3.52$|$0.29$|$7.19$|$0.11$|$1.17$|$\textbf{0.20}$|$3.72$|$0.09$|$1.80$|$0.15$|$\textbf{1.70}$|$0.07$|$2.75$|
|$\textbf{Ours}$|$\textbf{0.19}$|$\textbf{2.98}$|$\textbf{0.28}$|$\textbf{4.79}$|$0.24$|$1.78$|$\textbf{0.20}$|$\textbf{2.52}$|$0.13$|$1.84$|$\textbf{0.14}$|$1.88$|$0.10$|$2.77$|

||$\text {Gargoyle}$ ||||$\text {Lord Quas}$ ||||
|:---:|:---:|:---:|:---:|:---:|:---:|:---:|:---:|:---:|
||$\text {GT}$||$\text{Scans}$||$\text{GT}$||$\text { Scans }$||
|$\text{Method}$|$d_C$|$d_H$|$d_{\vec{C}}$|$d_{\vec{H}}$|$d_C$|$d_H$|$d_{\vec{C}}$|$d_{\vec{H}}$|
|$\text{DiGS}$|$\textbf{0.17}$|$\textbf{4.10}$|$0.09$|$0.92$|$\textbf{0.12}$|$\textbf{0.91}$|$0.06$|$0.70$|
|$\textbf{Ours}$|$0.19$|$4.56$|$0.15$|$1.82$|$0.14$|$1.13$|$0.09$|$0.95$|

$\quad\quad\quad\quad\quad\quad\quad\quad\quad\quad\quad\quad$ Table 2: Comparison under Surface Reconstruction Benchmark.

- We explore the effects of different sampling strategies under SRB and the evaluation settings of DiGS. Note that Bbox-sampling means we sample randomly in the bounding box ([-1, 1]) of the input and then utilize them for our alignment term, and $k$ means the $k-th$ of the nearest sampling following IGR[ICLR 2020]:

||$\textbf{Mean}$||
|:---|:---:|:---:|
||$\textbf{GT}$||
|$\text{Method}$|$d_C$|$d_H$|
|$\text{Bbox-sampling}$|$\text{0.32}$|$\text{4.51}$|
|$\text{k=1}$|$\text{0.32}$|$\text{4.90}$|
|$\text{k=25}$|$\text{0.28}$|$5.87$|
|$\mathbf{k=50 (ours)}$| $\textbf{0.19}$|$\textbf{2.98}$|
|$\text{k=75}$|$\text{0.26}$|$\text{4.31}$|
|$\text{k=100}$|$\text{0.28}$|$\text{5.34}$|

$\quad\quad\quad\quad\quad\quad\quad\quad\quad\quad\quad\quad$ Table 3: Effects of different sampling strategies.

---
Please let us know if you have any other questions. We would be pleased to engage in further dialogue with you.

Thanks for your time,

The Authors

---

### Comment · Area_Chair_Bags · 2023-08-12
**Author-reviewer discussion starts**

Dear reviewers,

Thanks for serving as reviewers.

The authors have submitted a rebuttal including a PDF with tables and figures. Please review through the rebuttal and reviews from other authors. If you have any questions, please feel free to let the authors know. You are more than welcome to post comments for further explanation or clarification before 1pm EDT on 8.21.

Best,
AC

---

### Comment · Area_Chair_Bags · 2023-08-18
**Kind reminders for reviewers**

Dear reviewers,

If you have not responded to the authors' feedback, please take some time to read through their responses and reviews from other reviewers. We would be very pleased to hear your thoughts.

Thanks,

AC

---

### Decision · Program_Chairs · 2023-09-21

**Decision:**

Accept (poster)

**Comment:**

This submission received 5 positive recommendations. Initially, the reviewers were concerned about the evaluations and the details. The authors addressed most of the concerns in the rebuttal and the responses to reviewers’ additional comments. The reviewers reached a consensus of acceptance after the discussion period. The AC read through the submission, the review, the rebuttal and the discussions. The AC agrees that the authors presented an interesting idea of constraining gradients in the learning of implicit representations, and also addressed the reviewers’ concerns by additional responses and convincing results. Per this, the AC supports the reviewers’ decision, and accepts this submission. The decision was discussed with and approved by the SAC. Please follow the reviewers’ comments to improve the manuscript.